# On-the-go Forgetting without Explicit Unlearning via ERASE

## Abstract

Existing unlearning approaches typically rely on post hoc weight adaptation or distillation, leading to duplicated memory costs, degraded generalization, and limited scalability. In this work, we introduce ERASE, *Erasure via Reconstructive Adversarial Signal Editing*, a framework for on-the-go forgetting that suppresses the observable influence of private data without modifying model weights. ERASE leverages structured, class-conditioned input perturbations to induce selective forgetting during inference, eliminating the need for retraining, fine-tuning, or model copies. We rigorously characterize sufficient conditions when ERASE provably achieves functional forgetting of designated subclasses while preserving predictions across other subclasses within the same superclass. This analysis offers a principled foundation for inference-time forgetting under mild regularity assumptions. Across diverse architectures and benchmark datasets, ERASE maintains the best observed balance between forgetting efficacy, computational efficiency, and retention fidelity over recent unlearning-based methods. By reimagining data removal as *forgetting without unlearning*, our work establishes a scalable, regulation-aligned pathway for continual, privacy-conscious learning.

## 1 Introduction

As machine learning systems increasingly interact with sensitive and continuously evolving data streams, their ability to *forget on demand* has become a cornerstone of privacy compliance and adaptive intelligence. In regulated domains such as healthcare, personalized services, and user-generated platforms, deployed models must often suppress the influence of outdated or private data (Li et al., 2025), not through retraining, but through immediate, behavioral forgetting. The challenge is to ensure that models no longer act on restricted data while maintaining utility on permissible data.

Conventional approaches to data removal primarily pursue *structural unlearning*, in which model parameters are explicitly modified to mimic the effect of retraining on a pruned dataset. Although this paradigm guarantees theoretical completeness, it is often computationally prohibitive and operationally impractical (Nguyen et al., 2022) in deployed systems that must respond to privacy or removal requests instantly. Such systems include on-device assistants that must forget on command, financial fraud detectors constrained by always-on operation, certified medical diagnostics that cannot be retrained after patient consent withdrawal, and live-service game AIs requiring instant behavioral updates.

We thus advocate a complementary notion, **functional forgetting**, that focuses not on erasing knowledge from weights, but on preventing its manifestation during inference. Functional forgetting treats data removal as a *behavioral* constraint rather than a retraining objective: the model retains its parameters intact yet behaves as if the forgotten data never existed when presented with relevant inputs. This perspective aligns with the needs of modern, continuously operating AI systems that must adapt in real time while preserving the stability of their learned representations.

To realize this notion, we propose ERASE (Erasure via Reconstructive Adversarial Signal Editing), a framework that operationalizes forgetting through **input-space modulation** rather than weight updates. ERASE leverages conditional priors and adversarial-style signal editing to selectively suppress responses associated with a designated forget set. By inducing erasure directly in the input domain, ERASE eliminates the need for retraining, fine-tuning, or access to internal gradients each time, thereby enabling *on-the-go compliance* in resource-constrained environments. Our key contributions are summarized as follows:

**1. On-the-go Forgetting Framework.** We propose ERASE, a novel framework for *inference-time forgetting* that enables models to suppress the influence of private data without retraining or gradient access each time. ERASE is designed for deployed settings, marking a shift from structural unlearning to functional forgetting.

**2. Selective Erasure via Conditional Adversarial Perturbations.** ERASE introduces *reconstructive adversarial signal editing*, a targeted input-space perturbation mechanism guided by *precomputed* conditional priors. This mechanism enforces forgetting only where necessary.

**3. Efficient and Deployable Privacy Compliance.** Our approach eliminates the need for retraining or auxiliary models, achieving extreme computational efficiency. It is well-suited for environments where model updates are impractical.

**4. Theoretical Insights.** We rigorously characterize sufficient conditions under which ERASE provably forgets the targeted subclass while retaining predictions on the other subclasses within the same superclass.

**5. Comprehensive Evaluation and Insights.** We evaluate ERASE across a wide range of model architectures and diverse datasets spanning multiple modalities. ERASE achieves competitive forgetting while substantially improving retention and deployment efficiency, even when compared against methods that rely on continual parameter updates. Ablation studies confirm that ERASE provides *targeted* forgetting.

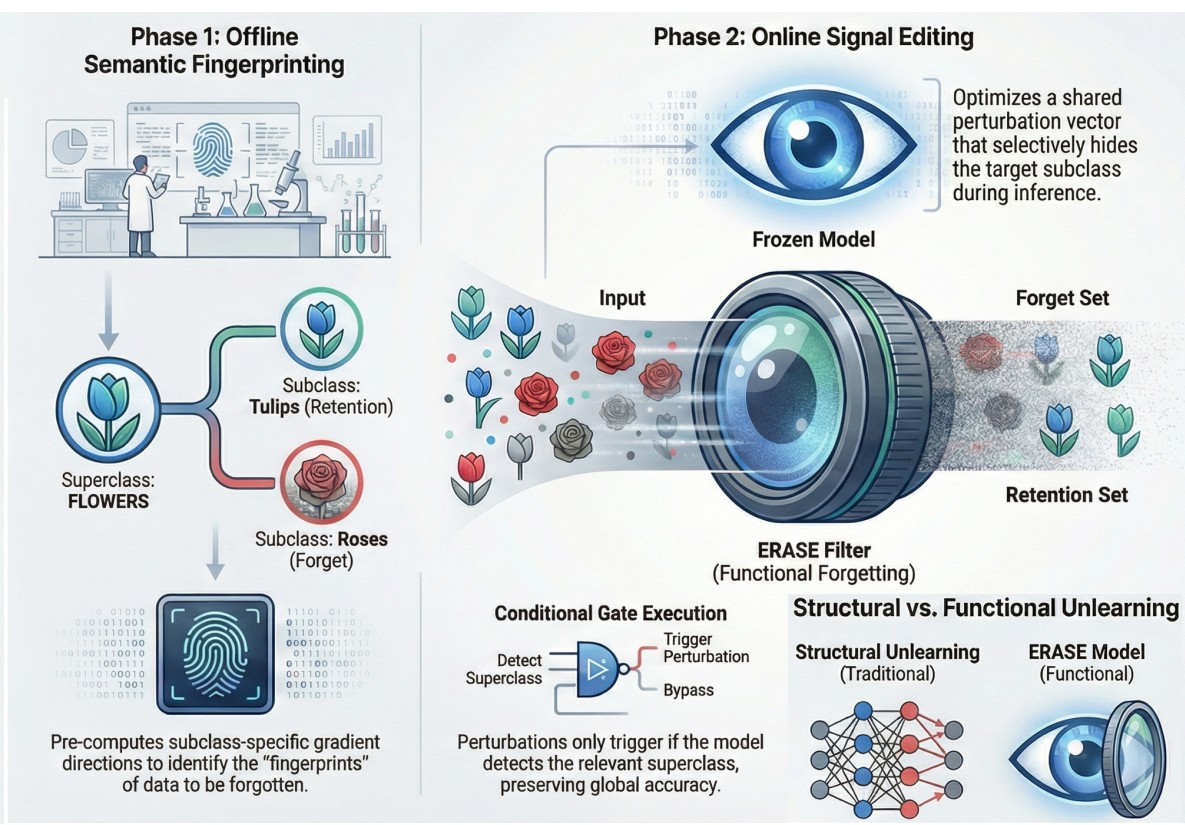

Figure 1: Overview of ERASE. In the offline phase, subclass-specific input-gradient priors are computed once from the frozen model and stored as semantic directions. In the online phase, these priors are combined into forget and retain directions, whose optimized combination is conditionally applied at inference time only when the input is predicted to belong to the target superclass. This suppresses the forgotten subclass while preserving retention on related sibling subclasses and the broader retain set, without fine-tuning the backbone model.

It is important to emphasize that our framework targets *functional forgetting* in discriminative models, where parameter modification is neither feasible nor desirable. While generative architectures may employ prompt-level filtering or guardrailing to achieve a similar goal, such mechanisms are inherently unavailable

in standard classification or recognition pipelines. `ERASE` is designed precisely for these deployment-level privacy scenarios, where the objective is to hide the effect of forgotten data rather than retrain it away.

## 2 Related work

Research on data removal in machine learning has been dominated by **machine unlearning**, where the goal is to eliminate the influence of designated training samples by *modifying model parameters* so that the resulting model approximates retraining on a pruned dataset. In this sense, most prior work targets what we refer to as **structural unlearning**: knowledge removal is achieved through updates to the model itself, whether via exact recomputation, approximate optimization, distillation, or selective retraining. This paradigm has produced important theoretical and practical advances, but it remains fundamentally tied to a *training-time* workflow and therefore incurs nontrivial computational, memory, or deployment overhead (Nguyen et al., 2022; Li et al., 2025). Such costs are particularly restrictive in always-on systems that must react to privacy, regulatory, or user-driven removal requests without retraining delays.

Our work is motivated by a different objective. Rather than enforcing deletion through parameter updates, we study **functional forgetting**, where the model parameters remain fixed and the desired forgetting behavior is induced only at inference time. The distinction is central: structural unlearning attempts to remove information from the weights, whereas functional forgetting aims to prevent that information from being expressed in downstream predictions. This framing is especially relevant for deployed discriminative models, where retraining, fine-tuning, or maintaining multiple model copies may be impractical. Within this broader landscape, the following categories of prior work illustrate the evolution of explicit unlearning approaches that our method conceptually extends and operationally supersedes.

**Exact and certified unlearning.** The earliest line of work sought strong deletion guarantees by making the post-unlearning model identical, or provably close, to a model retrained from scratch on the retained data. Such guarantees are achievable only for restricted model families or carefully designed pipelines, including settings such as linear and logistic regression, $k$-means, and random forests (Ginart et al., 2019; Mahadevan & Mathioudakis, 2022). Certified data removal methods further provide provable bounds on the deviation from exact retraining (Guo et al., 2020). These methods establish an important foundation for verifiable forgetting, but their assumptions and computational structure limit their applicability to modern deep models and large-scale deployment scenarios.

**Efficient retraining through partitioning and aggregation.** A second line of work improves the efficiency of structural unlearning by changing how the model is trained in the first place. Sharded, Isolated, Sliced, and Aggregated (SISA) training (Bourtoule et al., 2021) partitions the dataset into shards, trains separate sub-models, and aggregates their predictions, so that forgetting only requires retraining the affected shard rather than the full system. This reduces the cost of deletion and is broadly model-agnostic, but it still relies on retraining after a deletion request and may degrade utility because partitioning weakens cross-shard information sharing.

**Approximate unlearning via influence reversal and optimization.** To scale unlearning beyond exact methods, many approaches estimate and reverse the contribution of forgotten samples through approximate optimization. Representative techniques include influence functions (Koh & Liang, 2017), Newton-style removal updates (Guo et al., 2020), and gradient-based forgetting objectives such as gradient ascent on the forget set (Golatkar et al., 2020; Graves et al., 2021). These methods are attractive because they can be adapted to deep networks and are often substantially faster than retraining from scratch. However, they still alter the model parameters, and their approximate nature can leave residual traces of the removed data, which may be detectable under privacy audits such as membership inference attacks (Shokri et al., 2017).

**Corrective fine-tuning and representation-level forgetting.** More recent structural unlearning methods perform targeted corrective updates to suppress forgotten information. Examples include amnesiac unlearning (Graves et al., 2021), which applies counteracting gradient information, and impairing-data approaches that fine-tune the model to obscure sensitive content (Baumhauer et al., 2022). Other methods explicitly manipulate the geometry of the learned predictor, for example by shifting the decision boundary away from forget samples (Chen et al., 2023), or by masking entangled components and constraining knowl-

edge transfer to reduce re-emergence of forgotten information (Lin et al., 2023). These approaches improve targeting relative to naive retraining, but they remain dependent on post hoc parameter updates and often introduce additional optimization complexity, auxiliary components, or verification challenges.

**Distillation-based and teacher-guided unlearning.** A related family of methods uses knowledge distillation to disentangle forgetting and retention objectives. For example, Chundawat et al. (2023) employs an incompetent teacher to induce forgetting through deliberately misleading supervision, while Zhou et al. (2025b) decouples distillation into separate forgetting and retention modules to reduce interference between the two objectives. Distillation-based approaches can be effective, but typically require additional teacher models, repeated training procedures, or auxiliary modules, which increases system complexity and resource.

**Input-space and perturbation-based forgetting/unlearning.** More recently, a closely related line of work has begun to explore forgetting through *input-space* interventions rather than direct weight updates. Peng et al. (2025) studied catastrophic unlearning through an adversarial generator–unlearner framework that synthesizes challenging mixup samples to regularize the unlearning process. In parallel, recent class-wise unlearning methods have shown that targeted or universal adversarial perturbations (UAP) can directly stimulate forgetting behavior at the input level, including UAP-based class-wise forgetting (Zhou et al., 2025a) and input-agnostic "forget vectors" that drive machine unlearning while keeping model weights fixed (Sun et al., 2024). These works are especially relevant to our setting because they move unlearning closer to adversarial input manipulation. Our `ERASE` framework is most closely aligned with this emerging perspective, but differs in several important ways. Specifically, `ERASE` performs *conditional inference-time* forgetting for deployed discriminative models and uses an explicit forget/retain decomposition to preserve sibling subclasses within the same superclass. It is also hierarchy-aware and deployment-oriented, since perturbations are activated only when the model predicts the relevant superclass. In addition, `ERASE` provides theoretical sufficient conditions characterizing when targeted forgetting and retention can be achieved simultaneously. Note that Peng et al. (2025) and Zhou et al. (2025a) are not direct baselines for our setting, as the former uses an adversarial generator-unlearner with synthesized mixup samples, while the latter studies UAP-based class-wise unlearning. ERASE instead performs conditional inference-time functional forgetting for deployed discriminative models, with an explicit forget-retain decomposition for preserving sibling subclasses. Unlike prior perturbation-based unlearning works centered primarily on image classification, `ERASE` is evaluated across multiple modalities, supporting its applicability beyond visual settings.

Recent LLM unlearning works such as (Li et al., 2024; Wang et al., 2025; Yuan et al., 2025) target generative next-token models, prompt-response behavior, or LLM representation steering, and are therefore outside our discriminative superclass–subclass classification setting.

Overall, existing literature has substantially improved the efficiency, selectivity, and verifiability of structural unlearning, and recent perturbation-based methods have begun to show that forgetting can be induced from input side. Our `ERASE` framework builds on this emerging direction and advances it toward conditional, deployable, inference-time functional forgetting without retraining, fine-tuning, or auxiliary model copies.

## 3 Notations and problem formulation

Let $M_\theta$ be a fixed pre-trained discriminative model with parameters $\theta$, trained on a labeled dataset $\mathcal{D}_{\text{train}}$. Each data point in $\mathcal{D}_{\text{train}}$ is annotated with a subclass label (e.g., *Persian cat*) and a corresponding superclass label (e.g., CAT). The label hierarchy is structured such that multiple fine-grained subclasses (e.g., *Persian cat*, *tabby*, *cougar*) are grouped under broader superclasses (e.g., CAT). We emphasize that the pre-trained model $M_\theta$ is trained to map each input $x$ to its associated superclass label only, without access to subclass-level supervision. Given a target subclass $C_f$ to be unlearned, we posit to construct a (super)class-specific input-space perturbation that, when applied at inference time, selectively degrades the model's performance on samples from the *forget set*, while maintaining high accuracy on all remaining subclasses that collectively form the *overall retain set*. This objective stands in contrast to traditional machine unlearning, which instead aims to *perturb* the model parameters $\theta$ so that samples from the target subclass $C_f$ are misclassified at the superclass level, while preserving the model's performance on all other subclasses. We also define the *retain-superclass set* to consist of subclasses that belong to the same superclass of $C_f$, allowing for a more

---

**Algorithm 1** ERASE

---

1: **Require:** Model $M_\theta$, data $\mathcal{D}_{\text{train}}, \mathcal{D}_{\text{test}}$, target subclass $C_f$, forgetting weights $\omega_u, \omega_r > 0$, learning rate $\eta$
2: **Return:** Forget accuracy $\mathcal{A}_{\text{forget}}$, retain-superclass accuracy $\mathcal{A}_{\text{retain-super}}$, retain-overall accuracy $\mathcal{A}_{\text{retain-overall}}$
3: **Phase 1: Offline Generation of Subclass-Specific Gradient w.r.t. Input (one-time pre-computation)**
4: Initialize map $\mathbf{P} \leftarrow \{\}$
5: **for** each subclass $C_i \in \mathcal{D}_{\text{train}}$ **do**
6:     $\mathcal{D}_i \leftarrow \{(x,y) \in \mathcal{D}_{\text{train}} \mid \text{subclass}(x) = C_i\}, \quad \boldsymbol{\delta}_i \leftarrow 0$
7:     **for** each sample $(x_j, y_j) \in \mathcal{D}_i$ **do**
8:         $g \leftarrow \nabla_{x_j} \mathcal{L}(M_\theta(x_j), y_j)$
9:         $\boldsymbol{\delta}_i \leftarrow \boldsymbol{\delta}_i + \text{sign}(g)$
10:     **end for**
11:     $\mathbf{P}[C_i] \leftarrow \boldsymbol{\delta}_i / |\mathcal{D}_i|$
12: **end for**
13: **Phase 2: Online Perturbation-Based Forgetting at Inference Time**
14:           ▷ *Step 1: Define Forget and Retain Directions*
15: $S_f \leftarrow \text{superclass of } C_f, \quad \mathcal{R}_r \leftarrow \{C_i \in S_f \mid C_i \neq C_f\}$
16: $\boldsymbol{\delta}_{\text{forget}} \leftarrow \mathbf{P}[C_f] / \|\mathbf{P}[C_f]\|_F$
17: $\boldsymbol{\delta}_{\text{retain}} \leftarrow \sum_{C_i \in \mathcal{R}_r} \mathbf{P}[C_i] / |\mathcal{R}_r|$
18: $\boldsymbol{\delta}_{\text{retain}} \leftarrow \boldsymbol{\delta}_{\text{retain}} / \|\boldsymbol{\delta}_{\text{retain}}\|_F$
19:           ▷ *Step 2: Optimize Perturbation Scaling Factors*
20: Initialize $\epsilon_f, \epsilon_r > 0$
21: $\mathcal{D}_{\text{opt}} \leftarrow \{(x, y_{\text{super}}) \in \mathcal{D}_{\text{train}} \mid \text{superclass}(x) = S_f\}$
22: **for** each optimization epoch **do**
23:     **for** batch $\{(x_j, y_j)\}_{j=1}^B$ in $\mathcal{D}_{\text{opt}}$ **do**
24:         $\boldsymbol{p} \leftarrow \epsilon_f \cdot \boldsymbol{\delta}_{\text{forget}} - \epsilon_r \cdot \boldsymbol{\delta}_{\text{retain}}$
25:         $x'_j \leftarrow x_j + \boldsymbol{p} \cdot \|x_j\|_F, \quad o'_j \leftarrow M_\theta(x'_j)$
26:         $u_j \leftarrow \mathbf{1}_{[\text{subclass}(x_j) = C_f]}$
27:         $L_{\text{opt}} \leftarrow \frac{1}{B} \sum_{j=1}^B (\omega_r(1 - u_j) - \omega_u u_j) \mathcal{L}(o'_j, y_j)$
28:         $(\epsilon_f, \epsilon_r) \leftarrow (\epsilon_f, \epsilon_r) - \eta \nabla_{(\epsilon_f, \epsilon_r)} L_{\text{opt}}$
29:     **end for**
30: **end for**
31: **Evaluation (Conditional Perturbation)**
32: $\boldsymbol{p}_{\text{final}} \leftarrow \epsilon_f \cdot \boldsymbol{\delta}_{\text{forget}} - \epsilon_r \cdot \boldsymbol{\delta}_{\text{retain}}$
33: Define $\text{Perturb}(x) = x + \boldsymbol{p}_{\text{final}} \|x\|_F$
34: Define $\text{ConditionalPerturb}(x) =$
35:     **if** $\arg\max M_\theta(x) = S_f$ **then return** $\text{Perturb}(x)$
36:     **else return** $x$
37: $\mathcal{D}_{\text{forget}} \leftarrow \{(x,y) \in \mathcal{D}_{\text{test}} \mid \text{subclass} = C_f\}$
38: $\mathcal{D}_{\text{retain-super}} \leftarrow \{(x,y) \mid \text{subclass} \in \mathcal{R}_r\}$
39: $\mathcal{D}_{\text{retain-overall}} \leftarrow \{(x,y) \mid \text{subclass} \neq C_f\}$
40: $\mathcal{A}_{\text{forget}} \leftarrow \text{Accuracy}(M_\theta, \mathcal{D}_{\text{forget}}, \text{ConditionalPerturb})$
41: $\mathcal{A}_{\text{retain-super}} \leftarrow \text{Accuracy}(M_\theta, \mathcal{D}_{\text{retain-super}}, \text{ConditionalPerturb})$
42: $\mathcal{A}_{\text{retain-overall}} \leftarrow \text{Accuracy}(M_\theta, \mathcal{D}_{\text{retain-overall}}, \text{ConditionalPerturb})$

---

nuanced assessment of forgetting within semantically similar concepts. Notably, the *retain-superclass set* is a subset of the *overall retain set*. For the test set $\mathcal{D}_{\text{test}}$, we define the following subsets for evaluation:

- $\mathcal{D}_{\text{forget}}$: test samples from target subclass $C_f$ (forget set);
- $\mathcal{D}_{\text{retain-overall}}$: test samples not belonging to $C_f$ (retain set);
- $\mathcal{D}_{\text{retain-super}}$: subset of $\mathcal{D}_{\text{retain-overall}}$ with samples from other subclasses in the same superclass of $C_f$.

This hierarchy formulation is intentionally fine-grained. Unlike standard class-level forgetting, ERASE asks the model to suppress one subclass while preserving sibling subclasses that share the same superclass decision boundary. $\mathcal{D}_{\text{forget}}$ and $\mathcal{D}_{\text{retain-overall}}$ form a partition of $\mathcal{D}_{\text{test}}$, while $\mathcal{D}_{\text{retain-super}} \subset \mathcal{D}_{\text{retain-overall}}$ enables us to assess retention performance on semantically adjacent concepts within the most closely related categories to the forgotten subclass. Our objective is to minimize accuracy on $\mathcal{D}_{\text{forget}}$ while minimizing degradation of accuracy on the retain sets $\mathcal{D}_{\text{retain-super}}$ and $\mathcal{D}_{\text{retain-overall}}$.

## 4 ERASE: Erasure via Reconstructive Adversarial Signal Editing

We now introduce ERASE, our novel framework for inference-time forgetting. Unlike existing approaches for discriminative models that rely on model retraining, ERASE achieves forgetting by strategically perturbing the model inputs during inference, thereby modifying its predicted output without altering its parameters.

## 4.1 Overview of `ERASE`

`ERASE` is a model update-free forgetting framework that operates purely at inference time, requiring no update of model parameters. It proceeds in two main phases: (i) an *offline phase* pre-computing subclass-specific gradient directions in the input space and storing them, followed by (ii) an *online phase* performing forgetting for a target subclass by applying an optimized superclass-specific perturbation vector to inference inputs. This perturbation is shared across all inputs from the same superclass and applied only if needed. Figure 1 illustrates the overall framework.

**Offline phase: Subclass-specific input-gradient direction computation.** We compute an average input-gradient direction for each subclass $C_i$ in the train set $\mathcal{D}_{\text{train}}$. We denote the subset of samples from subclass $C_i$ as

$$\mathcal{D}_i = \{(x, y_{\text{super}}) \in \mathcal{D}_{\text{train}} \mid \text{subclass of } x \text{ is } C_i\},$$

where $y_{\text{super}}$ denotes the true superclass label of sample $x$. The average input-gradient direction, specific to each $C_i$, is computed using a Fast Gradient Sign Method (FGSM)-style formulation (Goodfellow et al., 2015), yielding a signed and averaged vector:

$$\mathbf{P}[C_i] = \frac{1}{|\mathcal{D}_i|} \sum_{(x_j, y_j) \in \mathcal{D}_i} \text{sign} \left( \nabla_{x_j} \mathcal{L}(M_\theta(x_j), y_j) \right), \tag{1}$$

where $\mathcal{L}$ is a supervised loss, e.g., cross-entropy. The precomputed signed gradients $\mathbf{P}$ act as semantic "fingerprints" for each subclass in input-space and are stored for downstream forgetting use. The subclass annotations $C_i$ in $\mathcal{D}_i$ are solely used to identify the target samples to be forgotten.

**Online phase: Targeted forgetting via optimized perturbations.** The online phase begins with the selection of a subclass $C_f$ to unlearn. We let $S_f$ denote the superclass to which the target subclass $C_f$ belongs, and $\mathcal{R}_r = \{C_i \in S_f \mid C_i \neq C_f\}$ the set of other subclasses in $S_f$. Our goal is to construct a perturbation $\boldsymbol{p}$ that selectively degrades model performance on $C_f$ (forget set), while preserving accuracy on $\mathcal{R}_r$ (retain set). This requires accessing the pre-computed input-gradients $\mathbf{P}$ from the offline phase. We define the following input-gradient directions.

• The forget direction as the normalized gradient for $C_f$:

$$\boldsymbol{\delta}_{\text{forget}} = \frac{\mathbf{P}[C_f]}{\|\mathbf{P}[C_f]\|_F};$$

• The retain direction as the average normalized gradient across all other subclasses $\mathcal{R}_r$ in $S_f$:

$$\boldsymbol{\delta}_{\text{retain}} = \sum_{C_i \in \mathcal{R}_r} \frac{\mathbf{P}[C_i]}{\|\sum_{C_j \in \mathcal{R}_r} \mathbf{P}[C_j]\|_F}.$$

Normalization ensures that each direction contributes based purely on its semantic alignment rather than raw gradient magnitude, mitigating bias from subclass size or gradient scales. This allows the final perturbation to reflect true directional intent rather than being skewed by disproportionately large gradients. We then construct the perturbation vector as a linear combination of these two directions:

$$\boldsymbol{p} = \epsilon_f \cdot \boldsymbol{\delta}_{\text{forget}} - \epsilon_r \cdot \boldsymbol{\delta}_{\text{retain}}, \tag{2}$$

where $\epsilon_f, \epsilon_r > 0$ are scaling factors optimized in the next step. The positive coefficient for $\boldsymbol{\delta}_{\text{forget}}$ pushes samples away from the target subclass representation, while the negative coefficient for $\boldsymbol{\delta}_{\text{retain}}$ pulls samples toward non-forgotten peer subclasses, improving retention. After scaling, this perturbation vector $\boldsymbol{p}$ will be applied later to each input $x$ belonging to $S_f$. Specifically, we scale the perturbation $\boldsymbol{p}$ for each input $x$ by its Frobenius norm $\|x\|_F$, ensuring that the perturbation magnitude adapts proportionally to the input's scale. This design maintains invariance to variations in pixel-level intensity across inputs, and is inspired by adversarial robustness literature, where perturbations are often norm-bounded relative to the input as $\|x' - x\|_p \leq \epsilon \|x\|_p$ to preserve perceptual consistency and prevent over-perturbation of low-magnitude

inputs (Carlini & Wagner, 2017; Moosavi-Dezfooli et al., 2016). For this, we define input perturbation function Perturb($\cdot$) as

$$\text{Perturb}(x) = x + \boldsymbol{p} \, \|x\|_F.$$

Since $\boldsymbol{\delta}_{\text{forget}}, \boldsymbol{\delta}_{\text{retain}}$ are pure directional, strength of $\boldsymbol{p}$ can be adjusted via $\epsilon_f, \epsilon_r$ in the optimization phase.

**Optimization of perturbation scaling factors.** To balance forgetting and retention, we optimize $(\epsilon_f, \epsilon_r)$ using a weighted loss on a subset of training data $\mathcal{D}_{\text{opt}} \subseteq \mathcal{D}_{\text{train}}$ consisting samples from superclass $S_f$. To achieve the balance, we maximize the loss on the forget training samples perturbed by $\boldsymbol{p}$ and minimize the loss on the remaining training samples in $\mathcal{D}_{\text{opt}}$ perturbed by $\boldsymbol{p}$. For each sample, we define a binary indicator $u_j = 1$ if $x_j$ belongs to the forget set $C_f$, and $u_j = 0$ otherwise. The optimization loss is:

$$L_{\text{opt}} = \tfrac{1}{|\mathcal{D}_{\text{opt}}|} \sum_{(x_j, y_j) \in \mathcal{D}_{\text{opt}}} (\omega_r(1 - u_j) - \omega_u u_j) \times \mathcal{L}(M_\theta(\text{Perturb}(x_j)), y_j), \tag{3}$$

where $\omega_u, \omega_r > 0$ are user-defined weights emphasizing forgetting and retention respectively. For instance, when the forget and retain training sets are significantly unbalanced, the weights $\omega_u, \omega_r$ can be adjusted to compensate for this imbalance, ensuring that both objectives contribute meaningfully to the loss in equation 3. The formulation in equation 3 encourages the model to perform poorly on $C_f$ via negative weighting while reinforcing correct predictions on other classes via positive weighting. Notably, no model parameters are updated and only the perturbation vector is learned.

**Inference-time evaluation.** The final step is evaluation of the unlearning effect on the three relevant subsets of the test set, defined earlier as $\mathcal{D}_{\text{forget}}$, $\mathcal{D}_{\text{retain-overall}}$, and $\mathcal{D}_{\text{retain-super}}$. For this evaluation, $M_\theta$'s accuracy on a dataset $\mathcal{D}$ under the input perturbation function Perturb($\cdot$) is defined as

$$\text{Accuracy}(M_\theta, \mathcal{D}, \text{Perturb}) = \tfrac{1}{|\mathcal{D}|} \sum_{(x, y_{\text{super}}) \in \mathcal{D}} \mathbf{1}_{[\arg\max M_\theta(\text{Perturb}(x)) = y_{\text{super}}]}.$$

For each test sample $x$, we adopt the following conditional perturbation strategy:

1. The model $M_\theta$ first predicts the superclass label $\hat{y}_{\text{super}}$ without any perturbation.
2. If the sample is predicted to belong to the target superclass $\hat{y}_{\text{super}} = S_f$, we apply the perturbation Perturb($x$) and re-infer the class.
3. If not, the model's original prediction is retained, i.e., no perturbation is applied.

This selective perturbation ensures that samples unrelated to the forget subclass remain untouched, preserving global model performance and ensuring that the perturbation effect is localized to the relevant semantic region. Successful unlearning corresponds to low accuracy on $\mathcal{D}_{\text{forget}}$, indicating successful forgetting; and high accuracy on $\mathcal{D}_{\text{retain-super}}$ and $\mathcal{D}_{\text{retain-overall}}$, indicating effective retention. Our proposed method is summarized in Algorithm 1.

**Broader applicability.** ERASE remains applicable even in the absence of an explicit hierarchy. Users can define custom semantic groupings based on domain knowledge or task-specific criteria, such as clustering visually or functionally related classes. For example, in Tiny ImageNet, we manually construct superclasses by clustering semantically related subclasses using the WordNet ontology. In real-world datasets like HAR, where no hierarchy exists, we define superclasses as activity types (e.g., walking) and subclasses as individual subjects performing those activities. The grouping should be semantically or task coherent and treated as part of the deployment design, guided by dataset ontology, domain knowledge, or task-specific structure. Poorly chosen groupings may make the forgetting task less meaningful or artificially easy, e.g., if unrelated subclasses are placed under the same superclass.

## 4.2 Computational efficiency

A key advantage of ERASE lies in its lightweight, inference-time deployment that avoids any retraining.

**Offline precomputation.** The offline phase of ERASE consists of computing input-gradient directions for each subclass. For a given input $x$, this requires a single backpropagation step to compute the gradient

$\nabla_x \mathcal{L}(M_\theta(x), y_{\text{super}})$. Computing gradients with respect to the input involves significantly fewer dimensions than computing gradients with respect to the model parameters. Since this operation is performed once per training sample and only aggregated per subclass, the overall cost remains linear in the size of the training set. Moreover, this cost is amortized over all future forgetting operations, as the same set of subclass-specific gradient directions can be reused.

**Online perturbation optimization.** The online phase only involves optimizing two scalar weights $(\epsilon_f, \epsilon_r)$ over the subset $\mathcal{D}_{\text{opt}}$ of training samples within the superclass $S_f$ of the forget subclass $C_f$. This results in a highly efficient, low-dimensional optimization problem, typically converging in fewer than 10 epochs. Crucially, no model gradients or weight updates are required, and the model $M_\theta$ remains frozen throughout. Compared to retraining-based or continual fine-tuning methods, ERASE thus reduces memory and compute overhead by several orders of magnitude.

**Inference overhead.** At deployment, the inference cost of ERASE is negligible. For any test input, the model first performs a standard forward pass to determine its predicted superclass $\hat{y}_{\text{super}}$. Only if $\hat{y}_{\text{super}} = S_f$, the precomputed perturbation is applied before a second forward pass.

### 4.3 Theoretical analysis

We rigorously analyze ERASE, characterizing sufficient conditions under which the method provably forgets $C_f$ while retaining non-forgotten subclasses in $\mathcal{R}_r$.

**Preliminaries.** Let $g(x) := \nabla_x \mathcal{L}(M_\theta(x), S_f)$ denote the gradient of the loss w.r.t. input $x$ for the target superclass $S_f$. Let $f_\theta(x) \in \mathbb{R}^{|\mathcal{S}|}$ denote the vector of logits produced by the model. ERASE applies perturbations only when the model predicts the target superclass. We define this *gate event* as $\mathcal{E}(x) := \{\arg\max_{S \in \mathcal{S}} f_\theta(x)_S = S_f\}$. For any input $x$ satisfying $\mathcal{E}(x)$, we define the *forget superclass margin* as the gap between the target superclass logit and the next best superclass: $m_f(x) := [f_\theta(x)]_{S_f} - \max_{S \neq S_f} [f_\theta(x)]_S$. Our analysis relies on the following assumptions.

**Assumption 4.1.** There exists $L > 0$ such that for any perturbation $\Delta = \boldsymbol{p}\|x\|_F$:

$$|\mathcal{L}(M_\theta(x + \Delta), S_f) - \mathcal{L}(M_\theta(x), S_f) - \langle g(x), \Delta \rangle| \leq \frac{L}{2}\|\Delta\|_2^2.$$

**Assumption 4.2.** For all $x$ satisfying $\mathcal{E}(x)$: $|\langle g(x), \boldsymbol{\delta}_{\text{forget}}\rangle| \leq B_f$ and $|\langle g(x), \boldsymbol{\delta}_{\text{retain}}\rangle| \leq B_r$, for some constants $B_f, B_r > 0$.

**Assumption 4.3.** Phase-1 exhibits strictly positive empirical alignment: $\hat{\alpha}_{f,n} := \frac{1}{n}\sum_{x \in D_{C_f}} \langle g(x), \boldsymbol{\delta}_{\text{forget}}\rangle \geq \mu_f$, and $\hat{\alpha}_{r,m} := \frac{1}{m}\sum_{x \in D_{R_r}} \langle g(x), \boldsymbol{\delta}_{\text{retain}}\rangle \geq \mu_r$, for some $\mu_f, \mu_r > 0$, where $\{x_i^{(f)}\}_{i=1}^n \sim D_{C_f}$ and $\{x_i^{(r)}\}_{i=1}^m \sim D_{R_r}$ are the Phase-1 datasets.

Let $P_{C_f}$ and $P_{R_r}$ denote distributions of forget and retain subclasses. Fix $\delta \in (0, 1)$ and set $\delta_f = \delta_r = \delta/2$. We define $\varepsilon_f(n) := B_f\sqrt{\frac{\log(2/\delta_f)}{2n}}$ and gate probabilities $p_{\mathcal{E},f} := \Pr(\mathcal{E}(X)|X \sim P_{C_f})$ (analogously $\varepsilon_r(m), p_{\mathcal{E},r}$). We define the population alignment lower bounds: $\alpha_f := \frac{\hat{\alpha}_{f,n} - B_f(1 - p_{\mathcal{E},f}) - \varepsilon_f(n)}{p_{\mathcal{E},f}}$, $\alpha_r := \frac{\hat{\alpha}_{r,m} - B_r(1 - p_{\mathcal{E},r}) - \varepsilon_r(m)}{p_{\mathcal{E},r}}$. We define the events $\mathcal{G}_f$ and $\mathcal{G}_r$ as the cases where the true conditional expected alignment exceeds $\alpha_f$ and $\alpha_r$ respectively. Let $K := |\mathcal{S}|$ denote the number of superclasses.

**Theorem 4.4.** *Let Assumptions 4.1-4.3 hold and the computed Phase-1 gradients satisfy $\hat{\alpha}_{f,n} > B_f(1 - p_{\mathcal{E},f}) + \varepsilon_f(n)$ and $\hat{\alpha}_{r,m} > B_r(1 - p_{\mathcal{E},r}) + \varepsilon_r(m)$. Then, with probability at least $1 - \delta$ over the Phase-1 datasets, the events $\mathcal{G}_f$ and $\mathcal{G}_r$ hold with strictly positive bounds $\alpha_f, \alpha_r > 0$. Conditioned on this event, the following guarantees apply simultaneously.*
*(i) Forgetting: For any $X \sim P_{C_f}$ satisfying $\mathcal{E}(X)$ and $\langle g(X), \boldsymbol{\delta}_{\text{forget}}\rangle \geq \tau_f$ with $0 < \tau_f < \alpha_f$, the prediction changes ($\arg\max M_\theta(Perturb(X)) \neq S_f$) whenever*

$$\mathcal{L}(M_\theta(X), S_f) + \|X\|_F(\epsilon_f \tau_f - \epsilon_r B_r) - \frac{L}{2}\|X\|_F^2\|\boldsymbol{p}_{\text{final}}\|_2^2 > \log K. \tag{4}$$

*(ii) Retention: For any $X \sim P_{R_r}$ satisfying $\mathcal{E}(X)$ and $\langle g(X), \boldsymbol{\delta}_{\text{retain}}\rangle \geq \tau_r$ with $0 < \tau_r < \alpha_r$, the prediction is preserved ($\arg\max M_\theta(Perturb(X)) = S_f$) whenever*

$$\mathcal{L}(M_\theta(X), S_f) - \|X\|_F(\epsilon_r \tau_r - \epsilon_f B_f) + \frac{L}{2}\|X\|_F^2\|\boldsymbol{p}_{\text{final}}\|_2^2 < \log 2. \tag{5}$$

*Moreover, the alignment conditions hold with*

$$\Pr(\langle g(X), \boldsymbol{\delta}_{\text{forget}}\rangle \geq \tau_f \mid X \sim P_{C_f}, \mathcal{E}(X)) \geq \frac{\alpha_f - \tau_f}{B_f - \tau_f},$$

$$\Pr(\langle g(X), \boldsymbol{\delta}_{\text{retain}}\rangle \geq \tau_r \mid X \sim P_{R_r}, \mathcal{E}(X)) \geq \frac{\alpha_r - \tau_r}{B_r - \tau_r}.$$

Theorem 4.4 establishes two critical conditions for ERASE. First, certified forgetting is feasible only when the subclass-specific gradient alignment $\hat{\alpha}$ is distinct enough to overpower worst-case noise from model errors. Second, the inequalities reveal a geometric trade-off mediated by the curvature $L$: the linear terms $(\epsilon_f, \epsilon_r)$ drive the necessary behavioral shift, while the quadratic penalty $\|\boldsymbol{p}_{\text{final}}\|_2^2$ limits perturbation magnitude. The forgetting condition is expressed through a cross-entropy threshold: pushing the post-perturbation loss above $\log K$ guarantees that $S_f$ cannot remain the argmax, while keeping the post-perturbation retain loss below $\log 2$ guarantees that $S_f$ remains the unique argmax. These thresholds are conservative but easy to verify, since $\mathcal{L}(M_\theta(X), S_f)$ is already computed during training/evaluation and $K$ is fixed by the number of superclasses. ERASE succeeds when the optimized scalings $(\epsilon_f, \epsilon_r)$ are large enough to raise the forget loss beyond the $\log K$ threshold, yet balanced enough to satisfy the retain-loss threshold (see Appendix C for proof details).

## 5 Experiments

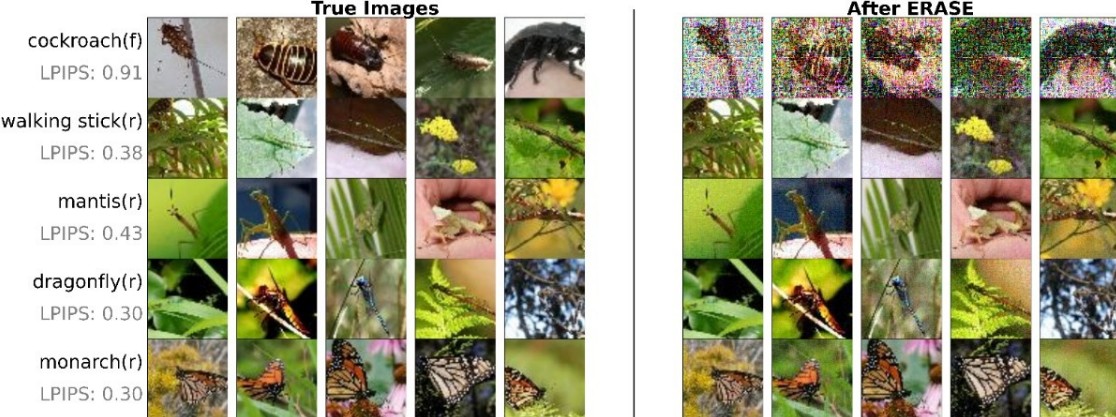

Figure 2: Selective corruption via ERASE on the INSECTS superclass containing the cockroach subclass in Tiny ImageNet. For each subclass, 5 representative test images are shown before (left) and after (right) perturbation. Cockroach images become noisier, disrupting predictions, while sibling subclasses remain largely unaffected (as also confirmed by LPIPS scores).

**Datasets and settings.** We benchmark ERASE across four diverse datasets, each reflecting different input modalities and subclass–superclass relationships:

- **CIFAR-100** (Krizhevsky et al., 2009): Comprises 100 fine-grained classes (subclasses) grouped into 20 coarse-grained categories (superclasses), as defined in the original dataset. We use ResNet18 (He et al., 2016) and ViT-B/16 (Dosovitskiy et al., 2021) as pre-trained backbone models for superclass-level classification.
- **Tiny ImageNet** mnmoustafa & Ali (2017): A subset of ImageNet with 200 subclasses. We manually cluster them into 40 semantic superclasses using WordNet hierarchies. ViT-B/16 serves as the backbone model.
- **Epileptic Seizure Recognition (ESZ)** (Shimanto, 2018): An EEG-based classification with 5 subclasses. Class 1 corresponds to subjects with epileptic seizures, while classes 2–5 are of non-seizure. These are grouped into two superclasses: *seizure* and *non-seizure*. A 3-layer feedforward neural network is employed.

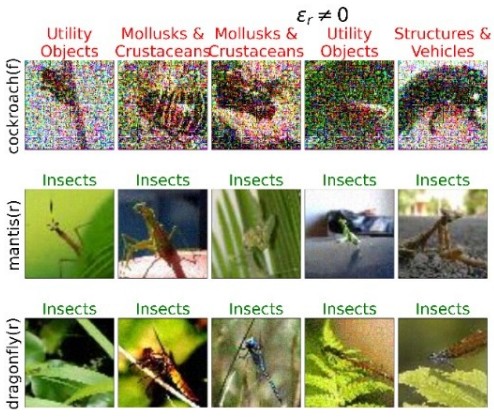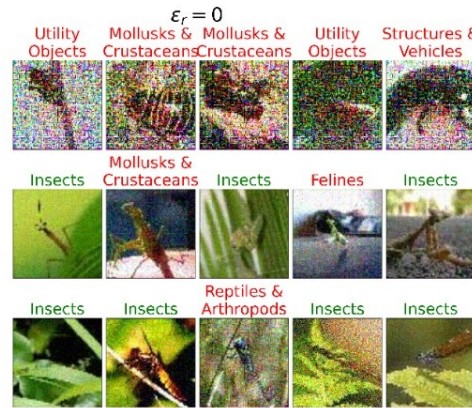

Figure 3: Effect of the retain direction $\boldsymbol{\delta}_{\text{retain}}$ in ERASE. With $\epsilon_r \neq 0$ (left), `cockroach` predictions are disrupted while retain classes are preserved. Without $\epsilon_r \neq 0$ (right), forgetting remains but collateral errors increase.

- **Human Activity Recognition (HAR)** (Anguita et al., 2013): A 6-class activity recognition task based on temporal sensor data from wearable devices. We treat each activity (e.g., walking, sitting) as a distinct superclass, with subclasses defined by individual subjects performing that activity. We use an LSTM-based sequence model (Hochreiter & Schmidhuber, 1997) for this task.

In all cases, models are trained to predict superclass labels. Subclass annotations are used exclusively for constructing the forget and retain sets.

**Baselines.** We compare ERASE with the following representative unlearning methods.

- **Gradient Ascent-Descent (GA+GD)** (Graves et al., 2021; Thudi et al., 2022): A gradient-based fine-tuning approach that maximizes loss on the forget set and minimizes it on the retain set.
- **Knowledge Distillation (KD)** (Chundawat et al., 2023): A recent approach employs a dual-teacher framework, where an *incompetent* teacher, trained to misclassify the forget set, induces forgetting, while a *competent* teacher, trained on the retain set, preserves relevant knowledge.
- **DEcoupLEd Distillation To Erase (DELETE)** Zhou et al. (2025b): A decoupled distillation framework that separates forgetting and retention, one module suppressing forget set representations and the other reinforcing retain set knowledge to avoid objective interference.
- **Boundary Unlearning** (Chen et al., 2023): A decision-boundary approach that shifts classifier boundaries away from forget samples while preserving retain class separation, achieved via local feature perturbations.
- **Entanglement-Reduced Mask & Knowledge Transfer and Prohibition (ERM-KTP)** (Lin et al., 2023): A feature space method that masks entangled neurons to suppress forget related activations and prohibits relearning during retraining for targeted, irreversible forgetting.

For each dataset, we fine-tune the publicly available pretrained model for 5 epochs to predict superclasses, and refer it as original model throughout the paper.

**Evaluation metrics.** We employ the following metrics:

- **Forget Accuracy** ($\mathcal{A}_{\text{forget}}$): Accuracy on test samples from the target forget subclass. Lower is better.
- **Retain Superclass Accuracy** ($\mathcal{A}_{\text{retain-super}}$): Accuracy on samples from sibling subclasses in the same superclass. Higher is better.
- **Retain Overall Accuracy** ($\mathcal{A}_{\text{retain-overall}}$): Accuracy on all other subclasses not equal to the target forget subclass. Higher is better.
- **Membership Inference Attack (MIA) Resistance** (Sidiropoulos et al., 2025): Computed on a balanced set of samples from the forget set (members) and the test set (non-members). A successful unlearning

should make these distributions indistinguishable, indicated by low Area Under the Curve (AUC-ROC) and a negative difference between the True Positive Rate (TPR) and False Positive Rate (FPR).

Since MIA primarily measures forgetting efficacy for privacy, we use it in conjunction with retention accuracy metrics. For ERASE, the attack evaluates the final model output $M_\theta(\text{ConditionalPerturb}(\cdot))$. We discuss additional privacy risks, including white-box limitations and timing side channels, in Appendix D.

| Method | Metric | CIFAR-100/ResNet18 | CIFAR-100/ViT | Tiny ImageNet | ESZ | HAR |
|---|---|---|---|---|---|---|
| Original Model | ↓ Forget | 82.83 ± 12.94 | 89.98 ± 9.71 | 83.47 ± 10.48 | 97.13 ± 4.45 | 97 ± 12 |
| | ↑ Retain Superclass | 82.83 ± 9.08 | 89.98 ± 5.83 | 83.47 ± 6.40 | 99.48 ± 0.29 | 96 ± 0 |
| | ↑ Retain Overall | 82.83 ± 0.13 | 89.98 ± 0.10 | 83.47 ± 0.05 | 97.13 ± 1.11 | – |
| Knowledge Distillation (Chundawat et al., 2023) | ↓ Forget | 27.39 ± 22.69 | 26.10 ± 25.99 | 14.32 ± 16.68 | 78.22 ± 37.96 | 87 ± 15 |
| | ↑ Retain Superclass | 77.41 ± 11.17 | 80.89 ± 10.31 | **70.93 ± 9.33** | 98.35 ± 0.92 | 95 ± 5 |
| | ↑ Retain Overall | 78.01 ± 0.59 | 81.27 ± 1.86 | 71.07 ± 1.32 | 95.99 ± 2.08 | – |
| GA+GD (Graves et al., 2021; Thudi et al., 2022) | ↓ Forget | 25.76 ± 20.24 | 24.52 ± 23.99 | 10.52 ± 2.94 | 72.57 ± 36.79 | 81 ± 30 |
| | ↑ Retain Superclass | **78.61 ± 8.81** | **83.98 ± 7.51** | 65.40 ± 12.71 | 97.33 ± 1.54 | **97 ± 7** |
| | ↑ Retain Overall | 81.24 ± 0.63 | 85.46 ± 1.41 | 67.45 ± 3.87 | 96.40 ± 1.83 | – |
| DELETE (Zhou et al., 2025b) | ↓ Forget | 18.41 ± 11.57 | **1.22 ± 3.08** | **0.01 ± 0.14** | **17.00 ± 10.15** | **3 ± 5** |
| | ↑ Retain Superclass | 40.57 ± 14.66 | 19.53 ± 20.95 | 9.24 ± 13.44 | 36.62 ± 31.77 | 82 ± 3 |
| | ↑ Retain Overall | 77.69 ± 2.6 | 52.00 ± 22.47 | 34.03 ± 28.20 | 51.77 ± 24.18 | – |
| Boundary Unlearning (Chen et al., 2023) | ↓ Forget | **9.85 ± 18.09** | 9.90 ± 15.82 | 4.59 ± 7.89 | 94.65 ± 6.06 | 89 ± 24 |
| | ↑ Retain Superclass | 26.61 ± 21.71 | 18.24 ± 21.40 | 40.45 ± 20.78 | 98.38 ± 0.91 | 96 ± 1 |
| | ↑ Retain Overall | 60.69 ± 5.54 | 34.37 ± 21.06 | 57.73 ± 21.50 | 96.76 ± 1.57 | – |
| ERM-KTP (Lin et al., 2023) | ↓ Forget | 16.19 ± 16.35 | 22.53 ± 21.06 | 15.48 ± 17.09 | 94.13 ± 9.65 | 92 ± 19 |
| | ↑ Retain Superclass | 25.84 ± 19.19 | 60.03 ± 20.05 | 55.53 ± 20.00 | **99.26 ± 0.43** | 95 ± 1 |
| | ↑ Retain Overall | 52.07 ± 11.09 | 59.32 ± 1.78 | 51.02 ± 2.08 | **97.88 ± 0.94** | – |
| ERASE (Proposed) | ↓ Forget | 28.53 ± 17.82 | 27.93 ± 18.86 | 14.64 ± 15.94 | 71.09 ± 36.08 | 76 ± 23 |
| | ↑ Retain Superclass | 76.01 ± 14.30 | 81.10 ± 9.66 | 68.80 ± 11.03 | 95.46 ± 2.64 | 93 ± 8 |
| | ↑ Retain Overall | **81.73 ± 0.44** | **88.73 ± 0.24** | **82.76 ± 0.20** | 94.12 ± 2.81 | – |

Table 1: Performance of ERASE and baseline unlearning methods across forgetting and retention accuracies.

| Method | Metric | CIFAR-100/ResNet18 | CIFAR-100/ViT | Tiny ImageNet | ESZ | HAR |
|---|---|---|---|---|---|---|
| Original Model | AUC-ROC | 0.98 | 0.98 | 0.96 | 0.63 | 0.62 |
| | (TPR, FPR, Δ) | (0.83, 0.03, 0.80) | (0.79, 0.03, 0.76) | (0.78, 0.04, 0.74) | (0.60, 0.30, 0.30) | (0.66, 0.43, 0.23) |
| Knowledge Distillation (Chundawat et al., 2023) | AUC-ROC | 0.43 | 0.57 | 0.42 | 0.35 | **0.35** |
| | (TPR, FPR, Δ) | (0.00, 0.17, **-0.17**) | (0.00, 0.01, -0.01) | (0.05, 0.15, -0.10) | (0.00, 0.14, -0.14) | (0.28, 0.53, **-0.25**) |
| GA+GD (Graves et al., 2021; Thudi et al., 2022) | AUC-ROC | 0.33 | **0.48** | 0.38 | 0.42 | 0.40 |
| | (TPR, FPR, Δ) | (0.05, 0.17, -0.12) | (0.07, 0.01, 0.06) | (0.04, 0.14, -0.10) | (0.20, 0.31, -0.11) | (0.38, 0.54, -0.16) |
| DELETE (Zhou et al., 2025b) | AUC-ROC | 0.38 | **0.48** | 0.30 | 0.47 | 0.41 |
| | (TPR, FPR, Δ) | (0.06, 0.16, -0.10) | (0.30, 0.17, 0.13) | (0.02, 0.15, **-0.13**) | (0.78, 0.78, 0.00) | (0.37, 0.53, -0.15) |
| Boundary Unlearning (Chen et al., 2023) | AUC-ROC | **0.19** | 0.49 | 0.32 | **0.32** | 0.40 |
| | (TPR, FPR, Δ) | (0.04, 0.16, -0.12) | (0.00, 0.02, **-0.02**) | (0.03, 0.16, **-0.13**) | (0.06, 0.19, -0.13) | (0.40, 0.57, -0.17) |
| ERM-KTP (Lin et al., 2023) | AUC-ROC | 0.50 | 0.49 | 0.46 | 0.34 | 0.47 |
| | (TPR, FPR, Δ) | (0.06, 0.16, -0.10) | (0.04, 0.02, 0.02) | (0.06, 0.16, -0.10) | (0.01, 0.17, **-0.16**) | (0.51, 0.60, -0.09) |
| ERASE (Proposed) | AUC-ROC | 0.37 | 0.50 | 0.40 | 0.40 | 0.37 |
| | (TPR, FPR, Δ) | (0.06, 0.16, -0.10) | (0.07, 0.02, 0.05) | (0.05, 0.15, -0.10) | (0.26, 0.39, -0.13) | (0.33, 0.52, -0.19) |

Table 2: Performance comparison of MIA vulnerability of unlearning methods. Lower AUC-ROC and Δ = TPR−FPR are better.

**Infrastructure & hyper-parameters.** Experiments were conducted on Kaggle with 2× NVIDIA Tesla T4 GPUs (15 GB each), AWS with 15 GB NVIDIA Tesla T4 GPUs, and Intel Xeon CPUs, with PyTorch 2.6.0+cu124. We adopted the hyperparameters of baseline methods from their respective publications, when available for certain datasets Chundawat et al. (2023); Zhou et al. (2025b); Chen et al. (2023); Lin et al. (2023). Our MIA employed a two-layer MLP attacker with a 64-neuron hidden layer (ReLU activation), which takes the target model's softmax output as input. Each attacker was trained for 10 epochs using Adam with learning rate 0.001. Hyperparameters of all unlearning methods are described in the supplementary text.

**Results.** Our experimental results, presented in Table 1 and Table 2, demonstrate that by decoupling the act of forgetting from model parameter modification and operating solely at inference time, ERASE not only achieves effective forgetting but, critically, preserves the model's vast repository of learned knowledge. This approach not only eliminates the computationally prohibitive overhead of retraining but also addresses the critical need for real-time privacy compliance.

• **Preservation of global knowledge.** The primary weakness of conventional unlearning methods is catastrophic forgetting, i.e., the unintended erasure of valuable, unrelated knowledge. Table 1 unequivocally

shows `ERASE`'s dominance in preventing this. On the large-scale Tiny ImageNet (ViT) dataset, `ERASE` achieves a *retain overall accuracy* of **82.76%**, nearly matching the original model (83.47%) and decisively outperforming all retraining-based competitors. Methods like DELETE (34.03%), Boundary Unlearning (57.73%), and GA+GD (67.45%) suffer from extensive collateral damage. Similarly, on CIFAR-100/ViT, `ERASE` sets a new SOTA with **88.73%** *retain overall accuracy*, proving its ability to surgically target information for forgetting. This performance is a direct result of our "forgetting without unlearning" approach–keeping the model frozen and conditionally perturbing the inputs.

- **Targeted forgetting.** While preserving knowledge is paramount, effective forgetting is the primary goal. `ERASE` demonstrates strong performance here as well, achieving a superior balance. Unlike methods such as DELETE and Boundary Unlearning, which achieve near-zero *forget accuracy* at the expense of severe model degradation (e.g., DELETE's 0.01% forget accuracy on Tiny ImageNet causes a 74% drop in superclass retention), `ERASE` achieves a competitive *forget accuracy* of 14.64% while keeping *retain superclass accuracy* high at 68.80%, comparable to KD (70.93%). This highlights `ERASE`'s ability to perform forgetting without disrupting closely related concepts. Furthermore, on real-world modalities ESZ and HAR, `ERASE` achieves second-best *forget accuracies* (**71.09%** and **76%** respectively), validating our input-space methodology's robustness beyond vision tasks.

- **A superior forgetting trade-off for deployed systems.** Across all datasets, `ERASE` consistently delivers the most practical and desirable trade-off between forgetting and retention. It avoids the false choice offered by other methods: either incomplete forgetting or a broken model.

- **Robustness to privacy attacks.** Beyond accuracy, we assess `ERASE`'s privacy guarantees through MIA, as summarized in Table 2. `ERASE` consistently demonstrates robust defense, achieving low AUC-ROC scores (e.g., 0.37 on CIFAR-100/ResNet18, 0.40 on Tiny ImageNet). Notably, its MIA resistance is on par with traditional weight-tuning methods, confirming that `ERASE` obscures the influence of forgotten data via input-space manipulation alone.

- **Ablation studies.** To understand the contribution of key components in `ERASE`, we conduct ablation experiment on Tiny ImageNet. Figure 2 visualizes the impact of our targeted input perturbation on the `cockroach` subclass within the INSECTS superclass. The figure shows representative test samples from all subclasses in INSECTS (5 images per subclass), before and after perturbation, i.e., $x$ and $\text{Perturb}(x)$. We observe that `cockroach` images become noticeably blurrier after perturbation, often disrupting the model's prediction. In contrast, sibling subclasses remain visually sharp and largely unaffected, showcasing `ERASE` preserves perceptual quality and predictive fidelity for the retain set $\mathcal{R}_r$, validating its targeted forgetting.

Figure 3 further elucidates the crucial role of the retain direction $\boldsymbol{\delta}_{\text{retain}}$. We compare predictions on INSECTS superclass from Tiny ImageNet with and without $\boldsymbol{\delta}_{\text{retain}}$ (i.e., $\epsilon_r \neq 0$ vs. $\epsilon_r = 0$). When $\epsilon_r \neq 0$, `ERASE` successfully degrades predictions for the forget class (top row), while preserving correct superclass predictions for retain subclasses (bottom rows). However, removing the retain term ($\epsilon_r = 0$) causes misclassifications for retain examples, e.g., `mantis` misclassified as FELINES, `dragonfly` as REPTILES & ARTHROPODS. It demonstrates $\boldsymbol{\delta}_{\text{retain}}$'s role in maintaining model utility on the retain set.

# 6    Conclusion

The dominant approach to data removal in machine learning has long relied on retraining or fine-tuning to achieve explicit unlearning. In this work, we challenged that paradigm and demonstrated that effective erasure does not require modifying a model's parameters. We introduced `ERASE`, a framework for on-the-go forgetting that operates at inference time. By reframing data deletion as a problem of behavioral suppression rather than structural modification, `ERASE` enables models to dynamically forget sensitive information while maintaining their original weights and predictive capability. This design delivers immediate, scalable, and regulation-compliant data removal without retraining overhead. Our experiments across diverse modalities show that `ERASE` achieves a superior balance between forgetting efficacy and retention fidelity, providing a principled pathway toward continual, privacy-conscious AI systems.

## Broader Impact Statement

This paper presents work whose goal is to advance the field of Machine Learning. There are many potential societal consequences of our work, none which we feel must be specifically highlighted here.

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

# A   Hyper-parameters details

Batch sizes were 64 (CIFAR, ESZ), 16 (Tiny ImageNet), and 128 (HAR). For the baseline methods, we adopted the hyperparameters from their respective publications when available (Chundawat et al., 2023; Zhou et al., 2025b; Chen et al., 2023; Lin et al., 2023).

Specifically, KD used Adam with learning rate 0.0001 (CIFAR, Tiny ImageNet), 0.01 (ESZ), and 0.001 (HAR); unlearning epoch was 1.

GA+GD used the same hyperparameters as KD.

DELETE used SGD with learning rate 0.001 (ResNet), 0.0003 (ViT), 0.03 (HAR) and momentum 0.9; Adam with learning rate 0.01 (ESZ); unlearning epochs were 2 (ResNet), 1 (ViT, ESZ), and 5 (HAR).

Boundary unlearning used adversarial noise bound 0.1; poison epochs 10; SGD with learning rate 0.00001 and momentum 0.9; curvature regularization coefficient 0.7; adjustment bias and slope in reweighting scheme as -0.5 and 5; step size for approximating the curvature 0.9.

ERM-KTP used distillation loss coefficient 0.1; forget set loss coefficient 1; temperature 2; unlearning epochs 1 (CIFAR, Tiny ImageNet), 5 (HAR), and 10 (ESZ); SGD with learning rate 0.01 (CIFAR) and 0.001 (Tiny ImageNet, ESZ, HAR), momentum 0.9, and weight decay 0.0005.

In ERASE, $\eta$ was set to 0.01 (CIFAR, ESZ) and 0.001 (Tiny ImageNet, HAR), selected from $0.1, 0.01, 0.001$; weights $(\omega_u, \omega_r) = (1.5, 0.8)$, chosen from $[1, 2] \times [0.5, 1]$, account for the smaller forget set.

Our MIA employed a two-layer MLP attacker with a 64-neuron hidden layer (ReLU activation), which takes the target model's softmax output as input. To ensure a robust evaluation, a distinct attacker, each trained for 10 epochs on a balanced dataset of 'member' and 'non-member' samples, was specialized for each subclass.

# B   Sensitivity to initialization and learning rate of $(\epsilon_f, \epsilon_r)$

We present sensitivity results on CIFAR-100/ResNet18 for two key optimization choices: initialization of $(\epsilon_f, \epsilon_r)$ and learning rate $\eta$, while keeping all other parameters fixed at their values mentioned above. The results in Tables 3-4 show that ERASE is relatively stable across these choices. Specifically, retain-overall accuracy remains around 81.5-81.7, while forget and retain-superclass accuracies vary moderately, reflecting the expected forgetting-retention trade-off. These additional results support that the $\epsilon$-optimization is not highly sensitive to reasonable choices of initialization or learning rate.

| $(\epsilon_{\mathbf{f0}}, \epsilon_{\mathbf{r0}})$ | Forget Acc. | Retain Superclass Acc. | Retain Overall Acc. |
|---|---|---|---|
| $(0.5, 0.5)$ | $28.53 \pm 17.82$ | $76.01 \pm 14.30$ | $81.73 \pm 0.44$ |
| $(0.2, 1.1)$ | $25.51 \pm 16.65$ | $74.03 \pm 13.89$ | $81.53 \pm 0.38$ |
| $(0.3, 0.7)$ | $28.27 \pm 17.64$ | $75.82 \pm 12.62$ | $81.64 \pm 0.30$ |
| $(0.3, 0.9)$ | $25.05 \pm 16.72$ | $74.96 \pm 13.35$ | $81.57 \pm 0.34$ |
| $(0.3, 1.0)$ | $25.55 \pm 16.46$ | $74.02 \pm 13.82$ | $81.53 \pm 0.37$ |
| $(0.4, 0.6)$ | $26.69 \pm 17.08$ | $75.46 \pm 13.15$ | $81.59 \pm 0.32$ |
| $(0.5, 0.6)$ | $26.54 \pm 15.54$ | $75.30 \pm 13.82$ | $81.50 \pm 0.36$ |

Table 3: Sensitivity to initialization of $(\epsilon_f, \epsilon_r)$ on CIFAR-100/ResNet18.

| $\eta$ | Forget Acc. | Retain Superclass Acc. | Retain Overall Acc. |
|---|---|---|---|
| 0.1 | $27.82 \pm 18.55$ | $74.90 \pm 15.01$ | $81.55 \pm 0.50$ |
| 0.01 | $28.53 \pm 17.82$ | $76.01 \pm 14.30$ | $81.73 \pm 0.44$ |
| 0.001 | $28.69 \pm 18.64$ | $75.13 \pm 15.75$ | $81.57 \pm 0.48$ |

Table 4: Sensitivity to learning rate $\eta$ on CIFAR-100/ResNet18.

## C   Theoretical analysis and proofs

In this section, we provide the complete theoretical analysis and rigorous proofs for the guarantees presented in the main paper. For completeness and to ensure this section is self-contained, we restate the necessary notations and formalize the assumptions introduced in Section 4 below, before presenting the detailed proof of Theorem 4.4.

### C.1   Preliminaries and notation

We consider a fixed, pre-trained classifier $f_\theta : \mathbb{R}^d \to \mathbb{R}^{|\mathcal{S}|}$, where $\mathcal{S}$ is the set of superclasses. For an input $x \in \mathbb{R}^d$, the output logits are denoted $f_\theta(x) = (f_\theta(x)_S)_{S \in \mathcal{S}}$. The predicted superclass is $\hat{S}(x) := \arg\max_{S \in \mathcal{S}} f_\theta(x)_S$.

We focus on a specific target superclass $S_f$. Within $S_f$, we designate a *forgotten subclass* $C_f$ and a set of *retained subclasses* $R_r = S_f \setminus \{C_f\}$. Let $P_{C_f}$ and $P_{R_r}$ denote the data distributions for the forgotten and retained subclasses, respectively. ERASE applies perturbations strictly conditionally. We define the gate event $\mathcal{E}(x)$ as the event where the model predicts the target superclass:

$$\mathcal{E}(x) := \{\hat{S}(x) = S_f\}. \tag{6}$$

If $\mathcal{E}(x)$ does not hold, the input is unmodified, and the original prediction is retained. Our analysis focuses on the non-trivial case where $\mathcal{E}(x)$ holds.

Let $\mathcal{L}(M_\theta(x), S_f)$ denote the cross-entropy loss with respect to the superclass $S_f$. We define the gradient of the loss with respect to the input as:

$$g(x) := \nabla_x \mathcal{L}(M_\theta(x), S_f). \tag{7}$$

We also define the *forget superclass margin $m_f(x)$*, which quantifies the confidence of the model in the target superclass $S_f$ relative to the next best superclass:

$$m_f(x) := f_\theta(x)_{S_f} - \max_{S \neq S_f} f_\theta(x)_S. \tag{8}$$

A negative margin $m_f(x) < 0$ implies $\hat{S}(x) \neq S_f$. Therefore, the condition for successful forgetting for an input $x \in C_f$ (where initially $\hat{S}(x) = S_f$) is to induce a perturbation $\Delta$ such that $m_f(x + \Delta) < 0$.

### C.2   Assumptions

We adopt the following assumptions (renumbered from Assumptions 4.1-4.3).

**Assumption C.1** (Directional Smoothness). The loss function $\mathcal{L}$ is smooth with respect to the input. Specifically, there exists a curvature constant $L > 0$ such that for any input $x$ and perturbation $\Delta = \boldsymbol{p}\|x\|_F$:

$$|\mathcal{L}(M_\theta(x + \Delta), S_f) - \mathcal{L}(M_\theta(x), S_f) - \langle g(x), \Delta \rangle| \leq \tfrac{L}{2}\|\Delta\|_2^2. \tag{9}$$

**Assumption C.2** (Bounded Projections). For all inputs $x$ satisfying the gate event $\mathcal{E}(x)$, the projection of the gradients onto the pre-computed directions is bounded:

$$|\langle g(x), \boldsymbol{\delta}_{\text{forget}} \rangle| \leq B_f, \quad \text{and} \quad |\langle g(x), \boldsymbol{\delta}_{\text{retain}} \rangle| \leq B_r, \tag{10}$$

for finite constants $B_f, B_r > 0$.

**Assumption C.3** (Empirical Alignment). The offline Phase-1 datasets $\mathcal{D}_{C_f} = \{x_i^{(f)}\}_{i=1}^n \sim P_{C_f}$ and $\mathcal{D}_{R_r} = \{x_j^{(r)}\}_{j=1}^m \sim P_{R_r}$ exhibit strictly positive empirical alignment with their respective computed directions:

$$\hat{\alpha}_{f,n} := \frac{1}{n}\sum_{i=1}^n \langle g(x_i^{(f)}), \boldsymbol{\delta}_{\text{forget}} \rangle \geq \mu_f > 0, \quad \hat{\alpha}_{r,m} := \frac{1}{m}\sum_{j=1}^m \langle g(x_j^{(r)}), \boldsymbol{\delta}_{\text{retain}} \rangle \geq \mu_r > 0. \tag{11}$$

## C.3 Proof of Theorem 1

Recall the following notation from Section 4.3. We let $P_{C_f}$ and $P_{R_r}$ denote the distributions of the forget and retain subclasses. We fixed $\delta \in (0, 1)$ and set $\delta_f = \delta_r = \delta/2$. We defined the gate event probabilities $p_{\mathcal{E},f} := \Pr(\mathcal{E}(X) \mid X \sim P_{C_f})$ and $p_{\mathcal{E},r} := \Pr(\mathcal{E}(X) \mid X \sim P_{R_r})$. To characterize the statistical error, let $\varepsilon_f(n) := B_f \sqrt{\frac{\log(2/\delta_f)}{2n}}$ and $\varepsilon_r(m) := B_r \sqrt{\frac{\log(2/\delta_r)}{2m}}$. We defined the population alignment lower bounds $\alpha_f$ and $\alpha_r$ as follows:

$$\alpha_f := \frac{\hat{\alpha}_{f,n} - B_f(1 - p_{\mathcal{E},f}) - \varepsilon_f(n)}{p_{\mathcal{E},f}},$$

$$\alpha_r := \frac{\hat{\alpha}_{r,m} - B_r(1 - p_{\mathcal{E},r}) - \varepsilon_r(m)}{p_{\mathcal{E},r}}.$$

We defined the alignment events:

$$\mathcal{G}_f := \{\mathbb{E}[\langle g(X), \boldsymbol{\delta}_{\text{forget}}\rangle \mid X \sim P_{C_f}, \mathcal{E}(X)] \geq \alpha_f\},$$
$$\mathcal{G}_r := \{\mathbb{E}[\langle g(X), \boldsymbol{\delta}_{\text{retain}}\rangle \mid X \sim P_{R_r}, \mathcal{E}(X)] \geq \alpha_r\}.$$

**Theorem C.4.** *Assume Assumptions C.1-C.3 hold and the computed Phase-1 gradients satisfy:*

$$\hat{\alpha}_{f,n} > B_f(1 - p_{\mathcal{E},f}) + \varepsilon_f(n) \quad and \quad \hat{\alpha}_{r,m} > B_r(1 - p_{\mathcal{E},r}) + \varepsilon_r(m).$$

*Let $K := |\mathcal{S}|$ denote the number of superclasses. Then, with probability at least $1 - \delta$ over the Phase-1 datasets, the events $\mathcal{G}_f$ and $\mathcal{G}_r$ hold with strictly positive bounds $\alpha_f, \alpha_r > 0$. Conditioned on this event, the following guarantees apply simultaneously:*

*(i) **Forgetting:** For any $X \sim P_{C_f}$ satisfying the gate event $\mathcal{E}(X)$ and the alignment condition $\langle g(X), \boldsymbol{\delta}_{\text{forget}}\rangle \geq \tau_f$ with $0 < \tau_f < \alpha_f$, the prediction changes $(\arg\max M_\theta(Perturb(X)) \neq S_f)$ whenever*

$$\mathcal{L}(M_\theta(X), S_f) + \|X\|_F(\epsilon_f\tau_f - \epsilon_r B_r) - \frac{L}{2}\|X\|_F^2\|\boldsymbol{p}_{\text{final}}\|_2^2 > \log K. \tag{12}$$

*(ii) **Retention:** For any $X \sim P_{R_r}$ satisfying the gate event $\mathcal{E}(X)$ and the alignment condition $\langle g(X), \boldsymbol{\delta}_{\text{retain}}\rangle \geq \tau_r$ with $0 < \tau_r < \alpha_r$, the prediction is preserved $(\arg\max M_\theta(Perturb(X)) = S_f)$ whenever*

$$\mathcal{L}(M_\theta(X), S_f) - \|X\|_F(\epsilon_r\tau_r - \epsilon_f B_f) + \frac{L}{2}\|X\|_F^2\|\boldsymbol{p}_{\text{final}}\|_2^2 < \log 2. \tag{13}$$

*Moreover, the probability that a random test sample satisfies the required alignment condition is lower bounded by:*

$$\Pr(\langle g(X), \boldsymbol{\delta}_{\text{forget}}\rangle \geq \tau_f \mid X \sim P_{C_f}, \mathcal{E}(X)) \geq \frac{\alpha_f - \tau_f}{B_f - \tau_f}, \tag{14}$$

$$\Pr(\langle g(X), \boldsymbol{\delta}_{\text{retain}}\rangle \geq \tau_r \mid X \sim P_{R_r}, \mathcal{E}(X)) \geq \frac{\alpha_r - \tau_r}{B_r - \tau_r}. \tag{15}$$

*Proof.* The proof proceeds in three stages: establishing the concentration of the pre-computed gradients to define strictly positive alignment bounds $\alpha_f, \alpha_r$ (Phase 1), deriving sufficient cross-entropy conditions for prediction change or preservation under perturbation (Phase 2), and lower bounding the probability of these conditions holding at test time.

**Step 1:** We first establish the existence of strictly positive lower bounds $\alpha_f$ and $\alpha_r$ for the conditional expected alignment. We focus on the forget set; the derivation for the retain set is identical.

Let $X \sim P_{C_f}$ and define the random variable $Z_f := \langle g(X), \boldsymbol{\delta}_{\text{forget}}\rangle$. By Assumption C.2, $Z_f \in [-B_f, B_f]$ almost surely. The Phase-1 dataset $\mathcal{D}_{C_f}$ consists of $n$ i.i.d. samples. Let $\hat{\alpha}_{f,n}$ be the empirical mean of $Z_f$ on $\mathcal{D}_{C_f}$. By Hoeffding's inequality, with probability at least $1 - \delta_f$:

$$\mathbb{E}[Z_f] \geq \hat{\alpha}_{f,n} - B_f\sqrt{\frac{\log(2/\delta_f)}{2n}} = \hat{\alpha}_{f,n} - \varepsilon_f(n). \tag{16}$$

The term $\mathbb{E}[Z_f]$ is the total expectation over $P_{C_f}$. We require the conditional expectation given the gate event $\mathcal{E}(X)$. Using the Law of Total Expectation:

$$\mathbb{E}[Z_f] = \mathbb{E}[Z_f \mid \mathcal{E}(X)]p_{\mathcal{E},f} + \mathbb{E}[Z_f \mid \mathcal{E}(X)^c](1 - p_{\mathcal{E},f}). \tag{17}$$

We bound the complement term using the worst-case magnitude: $\mathbb{E}[Z_f \mid \mathcal{E}(X)^c] \leq B_f$. Substituting this into the total expectation equation:

$$\mathbb{E}[Z_f] \leq \mathbb{E}[Z_f \mid \mathcal{E}(X)]p_{\mathcal{E},f} + B_f(1 - p_{\mathcal{E},f}). \tag{18}$$

Rearranging to solve for the conditional expectation and substituting the Hoeffding lower bound for $\mathbb{E}[Z_f]$:

$$\mathbb{E}[Z_f \mid \mathcal{E}(X)] \geq \frac{(\hat{\alpha}_{f,n} - \varepsilon_f(n)) - B_f(1 - p_{\mathcal{E},f})}{p_{\mathcal{E},f}}. \tag{19}$$

The RHS is exactly the definition of $\alpha_f$. By the condition $\hat{\alpha}_{f,n} > B_f(1 - p_{\mathcal{E},f}) + \varepsilon_f(n)$, the numerator is strictly positive, ensuring $\alpha_f > 0$. Thus, the event $\mathcal{G}_f := \{\mathbb{E}[Z_f \mid \mathcal{E}(X)] \geq \alpha_f\}$ holds with probability at least $1 - \delta_f$. An identical derivation applies to the retain set, yielding $\alpha_r > 0$ with probability $1 - \delta_r$.

Since the Phase-1 datasets $\mathcal{D}_{C_f}$ and $\mathcal{D}_{R_r}$ are drawn independently, we apply a union bound to ensure both bounds hold simultaneously:

$$\Pr(\mathcal{G}_f \cap \mathcal{G}_r) = 1 - \Pr(\mathcal{G}_f^c \cup \mathcal{G}_r^c) \geq 1 - (\Pr(\mathcal{G}_f^c) + \Pr(\mathcal{G}_r^c)) \geq 1 - (\delta_f + \delta_r) = 1 - \delta.$$

**Step 2:** Recall the perturbation $\Delta = \|X\|_F(\epsilon_f \boldsymbol{\delta}_{\text{forget}} - \epsilon_r \boldsymbol{\delta}_{\text{retain}})$, and let $\boldsymbol{p}_{\text{final}} = \epsilon_f \boldsymbol{\delta}_{\text{forget}} - \epsilon_r \boldsymbol{\delta}_{\text{retain}}$.

We first present a simple cross-entropy-to-argmax argument. For logits $z = f_\theta(x)$ and target superclass $S_f$, the softmax probability of $S_f$ is

$$p_{S_f} = \frac{\exp(z_{S_f})}{\sum_{S \in \mathcal{S}} \exp(z_S)}.$$

Therefore, the cross-entropy loss with target $S_f$ is

$$\ell(z, S_f) = \log\left[\sum_{S \in \mathcal{S}} \exp(z_S)\right] - z_{S_f}. \tag{20}$$

Factoring out $\exp(z_{S_f})$ from the log-sum-exp term gives

$$\sum_{S \in \mathcal{S}} \exp(z_S) = \exp(z_{S_f})\left(1 + \sum_{S \neq S_f} \exp(z_S - z_{S_f})\right).$$

Substituting this identity into the previous display yields

$$\ell(z, S_f) = \log\left[\exp(z_{S_f})\left(1 + \sum_{S \neq S_f} \exp(z_S - z_{S_f})\right)\right] - z_{S_f} = \log\left(1 + \sum_{S \neq S_f} \exp(z_S - z_{S_f})\right). \tag{21}$$

If $S_f$ remains the argmax among $K = |\mathcal{S}|$ superclasses, then its softmax probability is at least $1/K$, and hence $\ell(z, S_f) \leq \log K$. Therefore, $\ell(z, S_f) > \log K$ is a sufficient condition for $S_f$ not to remain the argmax. Similarly, if $\ell(z, S_f) < \log 2$, then the softmax probability of $S_f$ is larger than $1/2$, which implies that $S_f$ is the unique argmax.

(i) **Forgetting Condition:** Consider $X \sim P_{C_f}$ satisfying $\mathcal{E}(X)$. By Assumption C.1 (Directional Smoothness):

$$\mathcal{L}(M_\theta(X + \Delta), S_f) - \mathcal{L}(M_\theta(X), S_f) \geq \langle g(X), \Delta \rangle - \frac{L}{2}\|\Delta\|_2^2. \tag{22}$$

Conditioned on the alignment $\langle g(X), \boldsymbol{\delta}_{\text{forget}} \rangle \geq \tau_f$ and the bounded projection $|\langle g(X), \boldsymbol{\delta}_{\text{retain}} \rangle| \leq B_r$:

$$\langle g(X), \Delta \rangle = \|X\|_F (\epsilon_f \langle g(X), \boldsymbol{\delta}_{\text{forget}} \rangle - \epsilon_r \langle g(X), \boldsymbol{\delta}_{\text{retain}} \rangle) \geq \|X\|_F (\epsilon_f \tau_f - \epsilon_r B_r). \tag{23}$$

Substituting into the smoothness bound:

$$\mathcal{L}(M_\theta(X + \Delta), S_f) - \mathcal{L}(M_\theta(X), S_f) \geq \|X\|_F (\epsilon_f \tau_f - \epsilon_r B_r) - \frac{L}{2} \|X\|_F^2 \|\boldsymbol{p}_{\text{final}}\|_2^2. \tag{24}$$

Thus,

$$\mathcal{L}(M_\theta(X + \Delta), S_f) \geq \mathcal{L}(M_\theta(X), S_f) + \|X\|_F (\epsilon_f \tau_f - \epsilon_r B_r) - \frac{L}{2} \|X\|_F^2 \|\boldsymbol{p}_{\text{final}}\|_2^2. \tag{25}$$

If the right-hand side exceeds $\log K$, then $\mathcal{L}(M_\theta(X + \Delta), S_f) > \log K$. By the argument earlier, $S_f$ cannot remain the argmax, ensuring

$$\arg\max M_\theta(\text{Perturb}(X)) \neq S_f.$$

**(ii) Retention Condition:** Consider $X \sim P_{R_r}$ satisfying $\mathcal{E}(X)$. Using the smoothness upper bound:

$$\mathcal{L}(M_\theta(X + \Delta), S_f) - \mathcal{L}(M_\theta(X), S_f) \leq \langle g(X), \Delta \rangle + \frac{L}{2} \|\Delta\|_2^2. \tag{26}$$

Conditioned on $\langle g(X), \boldsymbol{\delta}_{\text{retain}} \rangle \geq \tau_r$ and bounding $|\langle g(X), \boldsymbol{\delta}_{\text{forget}} \rangle| \leq B_f$:

$$\langle g(X), \Delta \rangle \leq \|X\|_F (\epsilon_f B_f - \epsilon_r \tau_r). \tag{27}$$

The perturbed loss is therefore upper bounded by:

$$\mathcal{L}(M_\theta(X + \Delta), S_f) \leq \mathcal{L}(M_\theta(X), S_f) - \|X\|_F (\epsilon_r \tau_r - \epsilon_f B_f) + \frac{L}{2} \|X\|_F^2 \|\boldsymbol{p}_{\text{final}}\|_2^2. \tag{28}$$

If this upper bound is strictly smaller than $\log 2$, then $\mathcal{L}(M_\theta(X + \Delta), S_f) < \log 2$. By the argument earlier, the softmax probability of $S_f$ is larger than $1/2$, so $S_f$ is the unique argmax, ensuring

$$\arg\max M_\theta(\text{Perturb}(X)) = S_f.$$

Note: We use the thresholds $\log K$ and $\log 2$ because they give simple, distribution-free sufficient conditions. $\ell(z, S_f) > \log K$ guarantees that $S_f$ cannot remain the argmax, while $\ell(z, S_f) < \log 2$ guarantees that $S_f$ is the unique argmax. These thresholds are conservative but easier to verify than tighter margin-dependent conditions.

**Step 3:** Let $Y$ be the random variable $Z_f \mid \mathcal{E}(X)$. From Step 1, $\mathbb{E}[Y] \geq \alpha_f$. By Assumption C.2, $Y \leq B_f$. Define $W = B_f - Y \geq 0$. Then $\mathbb{E}[W] \leq B_f - \alpha_f$. By Markov's inequality:

$$\Pr(Y \geq \tau_f) = \Pr(B_f - W \geq \tau_f) = 1 - \Pr(W > B_f - \tau_f) \geq 1 - \frac{B_f - \alpha_f}{B_f - \tau_f} = \frac{\alpha_f - \tau_f}{B_f - \tau_f}. \tag{29}$$

This bound is valid and non-vacuous since $\alpha_f > 0$ (from Step 1) and we select $\tau_f < \alpha_f$. An identical bound applies to the retain set. $\square$

## C.4 Feasibility region of $(\epsilon_f, \epsilon_r)$

To explicitly characterize the range of $(\epsilon_f, \epsilon_r)$ under which both forgetting and retention are guaranteed, we analyze the geometry of the perturbation vector. This analysis formalizes the implicit trade-off between first-order directional alignment and second-order curvature penalties.

Starting from the two conditions 12-13 in Theorem C.4, forgetting requires

$$\mathcal{L}(M_\theta(X), S_f) + \|X\|_F (\epsilon_f \tau_f - \epsilon_r B_r) - \frac{L}{2} \|X\|_F^2 \|\boldsymbol{p}_{\text{final}}\|_2^2 > \log K,$$

while retention requires

$$\mathcal{L}(M_\theta(X), S_f) - \|X\|_F(\epsilon_r \tau_r - \epsilon_f B_f) + \frac{L}{2}\|X\|_F^2\|\boldsymbol{p}_{\text{final}}\|_2^2 < \log 2,$$

where $\boldsymbol{p}_{\text{final}} = \epsilon_f \boldsymbol{\delta}_{\text{forget}} - \epsilon_r \boldsymbol{\delta}_{\text{retain}}$ and $\rho := \langle \boldsymbol{\delta}_{\text{forget}}, \boldsymbol{\delta}_{\text{retain}} \rangle$. Since $\boldsymbol{\delta}_{\text{forget}}, \boldsymbol{\delta}_{\text{retain}}$ are unit vectors, expanding the squared Euclidean norm yields

$$\|\boldsymbol{p}_{\text{final}}\|_2^2 = \epsilon_f^2 + \epsilon_r^2 - 2\rho\,\epsilon_f\epsilon_r.$$

Dividing both inequalities by $\|X\|_F > 0$ and rearranging yields

$$\epsilon_f \tau_f - \epsilon_r B_r - \frac{\log K - \mathcal{L}(M_\theta(X), S_f)}{\|X\|_F} > \frac{L}{2}\|X\|_F(\epsilon_f^2 + \epsilon_r^2 - 2\rho\,\epsilon_f\epsilon_r),$$

$$\epsilon_r \tau_r - \epsilon_f B_f - \frac{\mathcal{L}(M_\theta(X), S_f) - \log 2}{\|X\|_F} > \frac{L}{2}\|X\|_F(\epsilon_f^2 + \epsilon_r^2 - 2\rho\,\epsilon_f\epsilon_r).$$

We formally define the feasible sets for each objective as:

$$F_f := \left\{ (\epsilon_f, \epsilon_r) : \epsilon_f \tau_f - \epsilon_r B_r - \frac{\log K - \mathcal{L}(M_\theta(X), S_f)}{\|X\|_F} > \frac{L}{2}\|X\|_F(\epsilon_f^2 + \epsilon_r^2 - 2\rho\epsilon_f\epsilon_r) \right\},$$

$$F_r := \left\{ (\epsilon_f, \epsilon_r) : \epsilon_r \tau_r - \epsilon_f B_f - \frac{\mathcal{L}(M_\theta(X), S_f) - \log 2}{\|X\|_F} > \frac{L}{2}\|X\|_F(\epsilon_f^2 + \epsilon_r^2 - 2\rho\epsilon_f\epsilon_r) \right\}.$$

The feasible region for simultaneously achieving forgetting and retention is the intersection

$$F = F_f \cap F_r.$$

Each set corresponds to a quadratic inequality in $(\epsilon_f, \epsilon_r)$. The left-hand side of each inequality is affine and represents the desired first-order effect of the perturbation, while the right-hand side is a quadratic curvature penalty that grows with the perturbation magnitude. This structure creates a bounded feasible region: the intersection $F$ contains coefficient pairs large enough to drive the intended behavior, i.e., pushing the post-perturbation forget loss above $\log K$ while keeping the post-perturbation retain loss below $\log 2$, yet small enough to avoid excessive second-order distortion.

The correlation term $\rho$ couples the two directions: when $\rho > 0$ (aligned directions), the quadratic penalty increases, shrinking $F$ and making simultaneous satisfaction harder. When $\rho < 0$ (opposing directions), the penalty weakens, enlarging $F$. In the special case $\rho = 0$ (orthogonal directions), the norm simplifies to $\|\boldsymbol{p}_{\text{final}}\|_2^2 = \epsilon_f^2 + \epsilon_r^2$. The region simplifies to:

$$F_f: \ \epsilon_f \tau_f - \epsilon_r B_r - \frac{\log K - \mathcal{L}(M_\theta(X), S_f)}{\|X\|_F} > \frac{L}{2}\|X\|_F(\epsilon_f^2 + \epsilon_r^2),$$

$$F_r: \ \epsilon_r \tau_r - \epsilon_f B_f - \frac{\mathcal{L}(M_\theta(X), S_f) - \log 2}{\|X\|_F} > \frac{L}{2}\|X\|_F(\epsilon_f^2 + \epsilon_r^2).$$

Here the geometric coupling disappears, and feasibility depends only on the total perturbation energy $\epsilon_f^2 + \epsilon_r^2$. This makes it easier to satisfy both guarantees when the forget and retain directions are nearly orthogonal, as optimizing one coefficient does not geometrically inflate the curvature cost of the other through a cross-term.

## D   Privacy and deployment risks

ERASE provides functional privacy in the deployment setting where the attacker observes the released inference behavior, e.g., the output of the prediction API after conditional perturbation. It should not be interpreted as structural removal of information from the model parameters. If an attacker obtains white-box access to the original model weights, then functional forgetting alone does not remove the information stored in those weights and does not preserve privacy against such an attacker. Similarly, the stored subclass directions and optimized perturbation coefficients are deployment artifacts that should be treated as sensitive and protected by access control.

Our MIA evaluation therefore measures black-box output-level membership leakage through softmax outputs after conditional perturbation. It does not prove complete removal of training influence. Stronger white-box, adaptive, influence-based, or reconstruction-style privacy audits test stronger threat models and remain important future work.

ERASE may also introduce a timing side channel. Since the conditional version may require a second forward pass when the gate predicts the target superclass $S_f$, an attacker observing latency could infer whether the gate was triggered. A simple mitigation is constant-time inference, e.g., always performing two forward passes or adding latency padding, so that gated and non-gated inputs are indistinguishable from timing.

