# OpenReview forum: "On-the-go Forgetting without Explicit Unlearning via ERASE"
_TMLR — Decision pending for TMLR_

### Review · Reviewer_gd3x · 2026-05-20

**Summary Of Contributions:**

This paper proposes ERASE, a “test-time unlearning” framework for classification models. Unlike traditional Machine Unlearning methods that achieve forgetting by updating model weights, ERASE keeps the model parameters unchanged and instead perturbs inputs belonging to target classes during inference by leveraging precomputed subclass-level input gradient directions, thereby reducing the prediction accuracy on forgotten subclasses.
The main contributions include:
1. Reformulating the unlearning problem as preventing the model from outputting hazardous information.
2. Conducting extensive experiments on datasets such as CIFAR-100, Tiny ImageNet, and ImageNet.
3. Proposing an input perturbation mechanism based on subclass-conditional gradient directions, including both forget directions and retain directions.

**Additional Comments:**

1. I am not sure if  some parts of this paper are AI-generated (e.g., Figure 1 and Algorithm 1).
2. The writing in Section 5 (Experiments) can be better organized.

**Audience:**

No

**Audience Explanation:**

1. First, I feel concerned about the motivation of this paper. The core idea is to learn a forgetting coefficient specifically for the target forget data. However, unlearning methods that do not modify model parameters inevitably rely on detecting the forgetting target itself. This raises another concern: does such target detection potentially constitute a different form of privacy violation?
1. I feel concerned that some claims in the paper are overstated. ERASE does not consistently achieve the best forgetting performance according to the experimental tables. Its main strength seems to be the trade-off between forgetting and retention, rather than superior forgetting itself.
1. I feel confused by the definition of “unlearning” used in this paper. Since ERASE does not modify model weights and only applies precomputed perturbations during inference, it is unclear whether the hazardous knowledge is actually removed. The method appears closer to inference-time suppression than genuine knowledge erasure.
1. I feel concerned that the experimental comparisons are outdated. Aside from DELETE, most baselines are relatively old and many recent machine unlearning methods are missing.
1. I feel doubtful about the practicality of the method. ERASE relies on precomputing and storing subclass-level gradient directions while assuming clear superclass–subclass structures. It is unclear how this assumption extends to open-category tasks, generative models, or large-scale real-world systems.

**Claims And Evidence:**

Yes

**Claims Explanation:**

The central claim of the paper is that ERASE can achieve forgetting without updating model weights by perturbing target subclass information at inference time, while still maintaining relatively high accuracy on the retain set. This claim is clearly defined in the paper and supported through algorithm descriptions and experiments. The authors explicitly evaluate the trade-off between forgetting and retention using metrics such as forget accuracy and retain accuracy. Overall, the experimental results support the authors’ claim that ERASE achieves strong retention performance. For example, on Tiny ImageNet, ERASE achieves a retain overall accuracy of 82.76%, which is close to the original model’s 83.47%.

**Requested Changes:**

1. Evaluate privacy more rigorously. The current MIA results only demonstrate weaker membership inference attacks on softmax outputs, but do not prove that training data influence has been removed. Stronger attacks, different attacker settings, and analyses on whether the perturbation merely hides output confidence rather than truly removing membership information should be included.
1. Add more recently published Machine Unlearning methods as baselines.
1. Include additional sensitivity analyses and ablation studies. The paper should systematically analyze the effects of superclass construction strategies, subclass counts, forget set size, ωu/ωr, η, ε initialization, perturbation norm constraints, and training data imbalance.
1. Analyze the behavior when the gating mechanism fails. ERASE only applies perturbation when the original model predicts the target superclass. If a forget sample is predicted into another superclass, or if a retain sample is mistakenly gated into the target superclass, how are forgetting and retention metrics affected? The paper would benefit from a more systematic error decomposition analysis.

---

> ### Author Response · Authors · 2026-05-29
> **Author Response (Part 1 of 3)**
>
> We thank the reviewer for the careful evaluation and for recognizing that ERASE’s central claim is clearly defined and supported by evidence. We appreciate the positive assessment of the retention evaluation. Below, we address the main concerns below.
>
> [**Re. Motivation and target detection**]: ERASE does not perform sample-level membership detection or infer whether an input was part of the training set. The forget target is specified externally by the user as a semantic subclass or concept, as in class-wise unlearning. ERASE stores only aggregate subclass-level gradient directions, not per-sample fingerprints or identifiers. At inference time, the gate uses only the model’s predicted superclass, i.e., information already produced by the classifier, and does not require access to private membership information. Thus, ERASE does not introduce an additional mechanism for detecting private training samples; it implements a specified concept-level forgetting request through functional suppression.
>
> [**Re. Overstated claims about forgetting performance**]: We agree that ERASE does not consistently achieve the lowest forget accuracy. Indeed, our intended claim is not that ERASE is uniformly best in forgetting alone, rather that it provides a favorable balance between forgetting, retention, and deployment efficiency, as stated in the Abstract, "Targeted forgetting" paragraph in Page 11, "A superior forgetting trade-off for deployed systems" paragraph in Page 12, and the Conclusion. Accordingly, in Contribution 5 on page 2, we have revised the phrase "ERASE achieves competitive, and in several cases state-of-the-art, forgetting performance,..." to "ERASE achieves competitive forgetting while substantially improving retention and deployment efficiency,...". Note that an unlearning method that simply randomizes or flips labels for all classes may achieve the strongest forgetting performance, but this typically comes at the cost of substantially degraded retention accuracy. Therefore, forgetting performance alone is not a sufficient metric for evaluating unlearning quality. An effective unlearning method should also preserve retain accuracy as much as possible. Our claims regarding ERASE are made precisely in this context.
>
>
> [**Re. Definition of unlearning vs inference-time suppression**]: We respectfully clarify that ERASE is not presented as structural, weight-level unlearning. The paper explicitly distinguishes structural unlearning from functional forgetting throughout the Title, Introduction, Related work, and method sections. ERASE is framed as "forgetting without explicit unlearning" where the model parameters remain fixed and the undesired subclass behavior is suppressed at inference time. Thus, the reviewer is correct that ERASE is inference-time behavioral suppression than certified weight erasure; however, this is precisely the problem setting and contribution of the paper, not an unintended ambiguity. To avoid any possible misunderstanding, we have further softened phrases such as "removes the influence" and "provably forgets" to "suppresses the observable influence" and "provably achieves functional forgetting".
>
> [**Re. Missing recent baselines**]: We used representative baselines for discriminative models, including gradient-based, distillation-based, boundary-based, and representation-level methods. There are several recent unlearning papers which are LLM-specific. We have added them to Related work, but not as experimental baselines because they target generative language models, next-token objectives, prompt-response behavior, or LLM representation steering, whereas ERASE is evaluated on discriminative superclass-subclass classification. Recent LLM-specific methods outside our experimental scope include [A] for Llama2-style generative models, [B] for hazardous-knowledge unlearning in LLMs, [C] for preference-style LLM unlearning, [D] for forget-dataset-only LLM unlearning, and [E] for objectives such as maximum-entropy and answer-preservation losses for LLM unlearning. These are important, but they are not directly comparable to our discriminative classifier setting.
>
> Note that, Peng et al. (2025) and Zhou et al. (2025a), despite being perturbation-based unlearning works, are not direct baselines for our setting. Peng et al. uses an adversarial generator-unlearner framework with synthesized mixup samples, while Zhou et al. focuses on UAP-based class-wise unlearning. ERASE instead performs conditional inference-time functional forgetting for deployed discriminative models, with an explicit forget-retain decomposition to preserve sibling subclasses within the same superclass, and is evaluated beyond image-only settings. We have made the aforementioned distinctions clearer to avoid suggesting that the recent related works were omitted from consideration.
>
> Continued below...

---

> ### Author Response · Authors · 2026-05-29
> **Author Response (Part 2 of 3)**
>
> ...Continued from above
>
> [A] Eldan, R., & Russinovich, M. (2023). Who’s harry potter? approximate unlearning in LLMs, arxiv. arXiv preprint arXiv:2310.02238.
>
> [B] Li, N., et al. (2024, July). The WMDP Benchmark: Measuring and Reducing Malicious Use with Unlearning. In International Conference on Machine Learning (pp. 28525-28550). PMLR.
>
> [C] Zhang, R., Lin, L., Bai, Y., & Mei, S. Negative Preference Optimization: From Catastrophic Collapse to Effective Unlearning. In First Conference on Language Modeling.
>
> [D] Wang, Y., et al. (2025, May). LLM unlearning via loss adjustment with only forget data. In International Conference on Learning Representations (Vol. 2025, pp. 43076-43104).
>
> [E] Yuan, X., Pang, T., Du, C., Chen, K., Zhang, W., & Lin, M. (2025, May). A closer look at machine unlearning for large language models. In International Conference on Learning Representations (Vol. 2025, pp. 49483-49508).
>
> [**Re. Practicality and superclass–subclass assumption**]: We respectfully clarify that the superclass–subclass structure is not merely an artificial assumption but part of the difficulty of our setting. ERASE targets subclass-level forgetting while preserving sibling subclasses within the same superclass. This is more challenging than standard class-level forgetting, because the forgotten and retained concepts are semantically adjacent and share the same superclass decision boundary. For instance, the goal may be to forget all cars while still correctly predicting other vehicle subtypes as vehicles, or to suppress the history/records of one subject while retaining those of other subjects in the same activity/category. The paper defines $\mathcal{D}_{\mathrm{retain-super}}$ precisely to evaluate this fine-grained retention behavior, and uses it throughout the experiments. We also note that ERASE does not require a canonical hierarchy: the paper states that custom semantic groupings can be defined using domain knowledge or task-specific criteria, as done with WordNet-based Tiny ImageNet groupings and HAR activity/subject groupings. We have strengthened the text to emphasize that our setting is intentionally fine-grained and more demanding than coarse class-level forgetting.
>
> We clarify that the scope is already stated at the end of Introduction: ERASE targets functional forgetting in discriminative classification models, not open-world generative unlearning. It explicitly contrasts ERASE with prompt-level filtering or guardrailing in generative architectures and states that ERASE addresses deployment-level scenarios where the goal is to hide the effect of forgotten data rather than retrain it away.
>
> [**Re. Stronger privacy evaluation**]: We agree that the current MIA does not prove complete removal of training influence. It evaluates black-box membership leakage through softmax outputs after conditional perturbation. However, this is appropriate for our stated deployment threat model: ERASE modifies what is exposed through the prediction API at inference-time. We have narrowed the privacy claim to black-box output-level MIA resistance. Stronger white-box, adaptive, influence-based, and reconstruction-style privacy audits are left for future work.
>
> [**Re. Additional sensitivity and ablation studies**]: We agree that a full sensitivity study over all these axes would be useful. Some factors are already addressed by design, reported in the paper, or varied across datasets. In particular, superclass construction is varied across datasets, including original hierarchies (CIFAR), WordNet-based grouping (Tiny ImageNet), and task-specific groupings (HAR). Subclass counts and forget-set sizes also vary naturally across the evaluated datasets/tasks. Perturbation scale is controlled through input-norm scaling and theoretically through the $\|p_{\mathrm{final}}\|_2^2$ curvature term. Training imbalance is handled through the weighted objective.
>
> To further address the reviewer’s concern, we added sensitivity results on CIFAR-100/ResNet18 for two key optimization choices: initialization of $(\epsilon_f,\epsilon_r)$ and learning rate $\eta$, while keeping all other parameters fixed as reported in the paper. The results show that ERASE is relatively stable across these choices: retain-overall accuracy remains around $81.5$--$81.7\%$, while forget and retain-superclass accuracies vary moderately, reflecting the expected forgetting-retention trade-off. These additional results support that the $\epsilon$-optimization is not highly sensitive to reasonable choices of initialization or learning rate.
>
> Continued below...

---

> ### Author Response · Authors · 2026-05-29
> **Author Response (Part 3 of 3)**
>
> ...Continued from above
>
> | $\mathbf{(\epsilon_{f0}, \epsilon_{r0})}$ | Forget Acc. ↓ | Retain Superclass Acc. ↑ | Retain Overall Acc. ↑ |
> |---:|---:|---:|---:|
> | (0.5, 0.5) | 28.53 ± 17.82 | 76.01 ± 14.30 | 81.73 ± 0.44 |
> | (0.2, 1.1) | 25.51 ± 16.65 | 74.03 ± 13.89 | 81.53 ± 0.38 |
> | (0.3, 0.7) | 28.27 ± 17.64 | 75.82 ± 12.62 | 81.64 ± 0.30 |
> | (0.3, 0.9) | 25.05 ± 16.72 | 74.96 ± 13.35 | 81.57 ± 0.34 |
> | (0.3, 1.0) | 25.55 ± 16.46 | 74.02 ± 13.82 | 81.53 ± 0.37 |
> | (0.4, 0.6) | 26.69 ± 17.08 | 75.46 ± 13.15 | 81.59 ± 0.32 |
> | (0.5, 0.6) | 26.54 ± 15.54 | 75.30 ± 13.82 | 81.50 ± 0.36 |
>
> | $\mathbf{\eta}$ | Forget Acc. ↓ | Retain Superclass Acc. ↑ | Retain Overall Acc. ↑ |
> |---:|---:|---:|---:|
> | 0.1 | 27.82 ± 18.55 | 74.90 ± 15.01 | 81.55 ± 0.50 |
> | 0.01 | 28.53 ± 17.82 | 76.01 ± 14.30 | 81.73 ± 0.44 |
> | 0.001 | 28.69 ± 18.64 | 75.13 ± 15.75 | 81.57 ± 0.48 |
>
> We have added these sensitivity results in the Appendix.
>
> [**Re. Gating failure analysis**]: Since ERASE applies perturbation conditionally only when the original prediction is $S_f$, all gate successes and failures are already included in the reported
> $\mathcal{A}_{\mathrm{forget}}$,
>
> $\mathcal{A}_{\mathrm{retain-super}}$,
>
> and $\mathcal{A}_{\mathrm{retain-overall}}$ metrics. Specifically, consider the following scenarios.
>
> 1. For a forget sample $x\in \mathcal{D}_{\mathrm{forget}}$, if the original model predicts a superclass other than $S_f$, then the sample is already misclassified and contributes zero to forget accuracy. Thus, not applying the perturbation in this case does not hurt the forgetting metric.
> 2. For a retain-overall sample outside the target superclass, e.g., a car sample when $S_f$ is insects, gate activation occurs only if the frozen model already wrongly predicts $S_f$. Thus, such a sample is already misclassified before ERASE. After perturbation, it may remain incorrect, change to another incorrect superclass, or occasionally become correct, but this case does not represent ERASE corrupting an originally correct outside-$S_f$ retain sample. In contrast, if an outside-$S_f$ retain sample is originally classified correctly, then its prediction is not $S_f$, the gate does not activate, and ERASE leaves it unchanged. Therefore, outside-superclass gate errors are reflected in $\mathcal{A}_{\mathrm{retain-overall}}$, but they are inherited from the learned model rather than introduced as new retention failures by ERASE.
>
> [**Re. Possible AI-generated Figure 1 / Algorithm 1**]: Algorithm 1 is author-written pseudocode, and not a single word is AI generated. Its algorithmic content directly summarizes the ERASE methodology presented in Section 4.1. For Figure 1, the conceptual blocks, texts, and schematic structure were author-designed; AI assistance was used only for presentation quality and visual refinement. Figure 1 is purely illustrative, contains no experimental result or data, and we verified its content. We strongly believe this use is consistent with TMLR’s LLM use policy for the authors.
>
> [**Re. Section 5 organization**]: We thank the reviewer for the suggestion. Section 5 already includes datasets/settings, baselines, metrics, hyperparameters, quantitative results, MIA evaluation, and ablations.

---

### Review · Reviewer_bgkR · 2026-05-22

**Summary Of Contributions:**

This paper introduces ERASE, a novel framework for achieving data forgetting at inference time without modifying the underlying model weights. By strategically perturbing input data based on pre-computed subclass-specific gradient directions, ERASE offers a computationally efficient and deployable alternative to traditional machine unlearning methods.

**Audience:**

Yes

**Audience Explanation:**

It's interesting know we can do unlearning in this way.

**Claims And Evidence:**

Yes

**Claims Explanation:**

Strengths:
1. ERASE completely bypasses the need for retraining, fine-tuning, or creating model copies. This drastically reduces computational costs, memory overhead, and latency.
2. The offline phase (gradient direction computation) is a one-time cost amortized over many inference operations.
3. The paper provides theoretical analysis that characterizes sufficient conditions for successful forgetting and retention. It establishes a trade-off between the linear effects of the perturbation and quadratic curvature penalties, offering insights into the feasibility and limits of the approach. (But some assumptions could be too strong which happen commonly in current papers.)
4. ERASE is evaluated across diverse datasets and modalities.

**Requested Changes:**

1. The core mechanism relies on a well-defined hierarchy of subclasses and superclasses. While the paper discusses adaptations for cases without explicit hierarchies (e.g., manual clustering, defining superclasses by activity types), the effectiveness heavily depends on the quality and appropriateness of this structure. A poorly defined hierarchy could lead to sub-optimal forgetting or retention.
2. While elegant, ERASE achieves "functional forgetting" (behavioral suppression) rather than "structural unlearning" (actual removal of knowledge from weights). This means the knowledge still exists within the model parameters; it's just being masked at inference time.
3. The core mechanism of generating input perturbations to control model output is inherently powerful but it could be repurposed as an "attack." The very directions that facilitate forgetting might exploit model weaknesses. While the paper's intent is unlearning, the underlying capability to manipulate predictions is shared with attack methodologies.
4. he paper primarily targets discriminative classification models. The "forget/retain direction" concept relies on gradient information derived from a specific loss function for class prediction.

---

> ### Author Response · Authors · 2026-05-29
> **Author Response (Part 1 of 1)**
>
> We thank the reviewer for the positive assessment of ERASE’s novelty, efficiency, and deployability. We appreciate the reviewer’s recognition of ERASE’s amortized offline cost, theoretical insight, and diverse evaluation. We address the reviewer’s requested clarifications below.
>
> [**Re. Hierarchy dependence**]: We respectfully clarify that the superclass--subclass structure is not merely an artificial assumption but part of the difficulty of our setting. ERASE targets subclass-level forgetting while preserving sibling subclasses within the same superclass. This is more challenging than standard class-level forgetting, because the forgotten and retained concepts are semantically adjacent and share the same superclass decision boundary. For instance, the goal may be to forget all cars while still correctly predicting other vehicle subtypes as vehicles, or to suppress the history/records of one subject while retaining those of other subjects in the same activity/category. The paper defines $\mathcal{D}_{\mathrm{retain-super}}$ precisely to evaluate this fine-grained retention behavior and uses it throughout the experiments. As you have identified, ERASE does not require a canonical hierarchy: the paper states that custom semantic groupings can be defined using domain knowledge or task-specific criteria, as done with WordNet-based Tiny ImageNet groupings and HAR activity/subject groupings. We will strengthen the text to emphasize that our setting is intentionally fine-grained and more demanding than coarse class-level forgetting.
>
> We agree that the quality of the grouping matters, but for the unlearning task. If the superclass is poorly constructed, the task may become either less meaningful or artificially easier. For example, grouping target subclass rabbit with retention subclasses car, truck, and bus would not test fine-grained retention among semantically adjacent concepts. ERASE is intended for groupings where the superclass captures a coherent semantic or task-specific relation, so that forgetting one subclass while retaining sibling subclasses is a meaningful and challenging objective. We have added this clarification and stated that the grouping should be treated as part of the deployment design, guided by dataset ontology, domain knowledge, or task-specific structure.
>
> [**Re. Functional forgetting vs structural unlearning**]: We agree with the reviewer’s characterization. However, this is precisely the problem setting and contribution of the paper, not an unintended ambiguity. ERASE is intentionally positioned as functional forgetting, not structural weight-level unlearning. The paper repeatedly states that ERASE keeps the model parameters fixed and suppresses the observable behavior of the forgotten subclass at inference time. This is not a limitation hidden by the paper; it is the core problem setting: deployment-time forgetting when retraining or parameter modification is infeasible or undesirable. To avoid any possible misunderstanding, we have further softened phrases such as "removes the influence" and "provably forgets" to "suppresses the observable influence" and "provably achieves functional forgetting under the stated conditions".
>
> [**Re. Possible misuse as an attack**]: We agree that input perturbations are a dual-use mechanism, since related techniques can be used adversarially. However, this concern is not unique to ERASE. Many unlearning baselines also intentionally induce selective forgetting by changing model parameters, shifting decision boundaries, or suppressing target representations. Such mechanisms could likewise be misused to create selectively failing or capability-degraded models. ERASE differs in purpose and deployment. Computing its perturbation directions require model weights access and are computed by the authorized model owner for a specified forgetting request.
>
> [**Re. Scope: discriminative classification models**]: We agree that ERASE targets discriminative classification models. This scope is already stated in the Introduction, where the paper contrasts ERASE with prompt-level filtering or guardrailing for generative architectures and states that ERASE addresses deployment-level scenarios where the goal is to hide the effect of forgotten data rather than retrain it away.
>
> [**Re. Some strong theoretical assumptions**]: We thank the reviewer for noting the theoretical analysis and for recognizing that such assumptions are common in concurrent work. We agree that our assumptions provide sufficient conditions rather than universal guarantees. Their role is to characterize when the forget and retain directions can simultaneously succeed, and to expose the trade-off between first-order directional alignment and second-order curvature penalties.

---

### Review · Reviewer_c7bp · 2026-05-25

**Summary Of Contributions:**

This paper introduces a novel inference time approach to forgetting classes. As opposed to conventional approaches that rely on structural unlearning, i.e., modifying model parameters to forget, this paper introduces a functional forgetting approach which suppresses specific behavior during inference, and hence, being to forget without explicit unlearning.

The authors present ERASE, a framework for on-the-go forgetting. It works in two phases:

- An offline phase that computes subclass-specific gradient directions in the input space.
- An online phase that performs forgetting by applying perturbation vectors to the inputs, calculated using the offline phase input-gradients.

The authors also provide a comprehensive theoretical analysis and demonstrate the efficacy of their method through empirical experiments across multiple datasets.

### **Strengths:**

- The paper is well-written, easy to follow, and supported by a good set of experiments and theoretical analysis.
- The approach is elegant, providing a fast, efficient, and simple way to perform on-the-go forgetting. I can see this being used effectively in practice.

### **Weaknesses:**

- In table 1 and table 2 of the paper, the best results should be highlighted in bold so that the tables are easier to read through.
- As this method is presented as something that can be deployed in actual privacy-preserving systems, I would encourage the authors to add a "Privacy and Performance Risks" section (at least in the appendix). This should briefly discuss how the system could be vulnerable to threats like timing attacks (since images must be passed twice during inference when the gate triggers) and white-box vulnerabilities (this technique will not preserve privacy if attackers gain direct access to the model weights), along with possible ways to make the system more secure.
- In the Theoretical Analysis (Appendix Section B), after equation 22 there’s a statement: “If the RHS exceeds the margin $m_f(X)$, the loss increase forces the superclass logit below a competing logit.” I am not sure if this statement is actually theoretically sound. The assumption that $\Delta \mathcal{L} > m_f(x)$ would guarantee a shift in the argmax is actually not guaranteed (although increased CE would imply sufficiently reduced confidence in $S_f$ which means the argmax likely shifts but it’s not guaranteed). I.e, Correct me if I am wrong but this can’t be claimed directly:

    $\Delta \mathcal L > m_f(x)
    \quad\Rightarrow\quad
    m_f(x+\delta)<0.$

    I think this can be resolved by introducing an explicit bound linking the CE Loss to the Logit Margin and then demonstrating that once the loss exceeds a specific threshold, the margin must become negative.

**Audience:**

Yes

**Audience Explanation:**

The method is novel and interesting and very effectively usable in production systems so I believe TMLR audiences would be interested in it

**Claims And Evidence:**

Yes

**Claims Explanation:**

The authors conduct a comprehensive set of empirical experiments as well as provide theoretical proofs for their method

**Requested Changes:**

Overall, I find the paper interesting, but I would encourage the authors to address the points mentioned in the weaknesses section.

---

> ### Author Response · Authors · 2026-05-29
> **Author Response (Part 1 of 1)**
>
> We thank the reviewer for the constructive and positive assessment. We appreciate that the reviewer found ERASE to be a clear, efficient, and practically relevant approach to inference-time forgetting, supported by both theoretical and empirical evidence. We address the requested clarifications below.
>
> [**Re. Table formatting**]: We thank the reviewer for the suggestion. We have highlighted in bold the best-performing entries in Tables 1 and 2 to improve readability.
>
> [**Re. Privacy and performance risks**]: We thank the reviewer for this useful suggestion. ERASE provides functional privacy only in the deployment setting where the attacker observes the released inference behavior, not the internal model. If an attacker obtains white-box access to the original model weights, then functional forgetting alone does not remove the information stored in the parameters and should not be interpreted as preserving privacy against such an attacker. Similarly, the stored subclass directions and optimized perturbation coefficients should be treated as sensitive deployment artifacts and protected by access control. So, practical safeguards include restricting access to model weights and stored directions.
>
> There is also a possible timing side channel: ERASE require a second forward pass when the gate predicts the target superclass $S_f$, so an attacker observing latency could infer whether the gate was triggered. A simple mitigation is to use constant-time inference, e.g., always perform two forward passes or add latency padding, so that gated and non-gated inputs are indistinguishable from timing. We have added a discussion in Appendix D.
>
> [**Re. Theoretical statement after Eq. 22**]: We thank the reviewer for this important observation. We have revised the proof by adding an explicit CE-loss-to-margin argument as follows.
>
> Let $K=|\mathcal{S}|$ be the number of superclasses, $z=f_\theta(X+\Delta)$ be the perturbed logits, and $m_f(z)=z_{S_f}-\max_{S\neq S_f}z_S$ be the post-perturbation margin. The cross-entropy loss with respect to $S_f$ can be written as $\ell(z,S_f)=\log\left(1+\sum_{S\neq S_f}\exp(z_S-z_{S_f})\right)$. If $S_f$ remains the argmax, then its softmax probability is at least $1/K$, and hence $\ell(z,S_f)\le \log K$. Equivalently, if $\ell(z,S_f)>\log K$, then $S_f$ cannot remain the argmax, so the prediction must change. Conversely, if $\ell(z,S_f)<\log 2$, then the softmax probability of $S_f$ is larger than $1/2$, so $S_f$ is the unique argmax, which gives a sufficient condition for retention.
>
> From Eq. 25 (earlier Eq. 22), define
>
> $A_f(X)=||X||_F(\epsilon_f\tau_f-\epsilon_rB_r)$
>
> $\hspace{5em}  -\frac{L}{2}$ $||X||_F^2$
>
> $\hspace{5em} \|p_{\mathrm{final}}\|_2^2$.
>
> Then $L(M_\theta(X+\Delta),S_f) \ge L(M_\theta(X),S_f)+A_f(X)$.
> Therefore, a valid sufficient condition for forgetting is
> $L(M_\theta(X),S_f)+A_f(X)>\log K$. Under this condition, the perturbed CE loss exceeds $\log K$, implying that the post-perturbation margin of $S_f$ is negative and hence $\arg\max M_\theta(X+\Delta)\neq S_f$. Similarly, for retention, the revised proof upper-bounds the perturbed CE loss and requires it to remain below $\log 2$. This guarantees that the softmax probability of $S_f$ is larger than $1/2$, and hence $\arg\max M_\theta(X+\Delta)=S_f$. Thus, both the forgetting and retention conditions can be expressed through calibrated CE-loss thresholds.
>
> These corrected conditions are also cleaner to verify in practice. $\mathcal{L}(M_\theta(X),S_f)$ is already computed during training/evaluation, while $\log K$ and $\log 2$ are fixed thresholds once the number of superclasses is known.

---

> > ### Comment · Reviewer_c7bp · 2026-06-14
> > **Response to Authors**
> >
> > Thank you for the detailed response and for updating the manuscript. I am satisfied with the changes.

---

> > > ### Author Response · Authors · 2026-06-14
> > > **Thank you for your feedback**
> > >
> > > We are glad that the revision has satisfactorily addressed your concerns. Thank you for your constructive feedback and continuous engagement throughout the process.

---

### Author Response · Authors · 2026-05-29
**Common Response to the Reviews**

We sincerely thank the Action Editor and the Reviewers for their detailed and constructive feedback. We have carefully considered all comments and addressed each of them in the detailed response provided below. Based on this feedback, we have revised the manuscript where appropriate and uploaded the revised version. We believe that these changes have improved the manuscript’s technical quality and readability. Please note that all revisions in the updated manuscript are highlighted in blue text. We thank the editorial team in advance for their time and consideration in reviewing our response and the resubmitted manuscript.